# Meta-analysis of gut microbiome studies identifies disease-specific and shared responses

Claire Duvallet[1,2], Sean M. Gibbons[1,2,3], Thomas Gurry[1,2,3], Rafael A. Irizarry[4,5] & Eric J. Alm[1,2,3]

Hundreds of clinical studies have demonstrated associations between the human microbiome and disease, yet fundamental questions remain on how we can generalize this knowledge. Results from individual studies can be inconsistent, and comparing published data is further complicated by a lack of standard processing and analysis methods. Here we introduce the MicrobiomeHD database, which includes 28 published case–control gut microbiome studies spanning ten diseases. We perform a cross-disease meta-analysis of these studies using standardized methods. We find consistent patterns characterizing disease-associated microbiome changes. Some diseases are associated with over 50 genera, while most show only 10–15 genus-level changes. Some diseases are marked by the presence of potentially pathogenic microbes, whereas others are characterized by a depletion of health-associated bacteria. Furthermore, we show that about half of genera associated with individual studies are bacteria that respond to more than one disease. Thus, many associations found in case–control studies are likely not disease-specific but rather part of a non-specific, shared response to health and disease.

[1] Department of Biological Engineering, Massachusetts Institute of Technology, Cambridge, MA 02139, USA. [2] Center for Microbiome Informatics and Therapeutics, Massachusetts Institute of Technology, Cambridge, MA 02139, USA. [3] The Broad Institute of MIT and Harvard, Cambridge, MA 02139, USA. [4] Department of Biostatistics and Computational Biology, Dana-Farber Cancer Institute, Boston, MA 02215, USA. [5] Department of Biostatistics, Harvard T.H. Chan School of Public Health, Boston, MA 02115, USA. Correspondence and requests for materials should be addressed to E.J.A. (email: ejalm@mit.edu)

The human gastrointestinal tract digests food, absorbs nutrients, and plays important roles in maintaining metabolic homeostasis. The microbes residing in our gut harvest energy from the food we eat, train our immune system, break down xenobiotics and other foreign products, and release metabolites and hormones important for regulating our physiology[1–3]. Chemical signals from our microbiota can act locally within the gut, and can also have larger systemic effects (e.g., the "gut-brain axis")[4–6].

Due to the physiological interplay between humans and our microbial communities, many diseases are hypothesized to be associated with shifts away from a "healthy" gut microbiome. These include metabolic disorders, inflammatory and auto-immune diseases, neurological conditions, and cancer, among others[1, 3, 7–9]. Certain gut-related conditions (e.g., obesity and inflammatory bowel disease) have been extensively studied in human cohorts and in animal experiments, where significant and sometimes causal microbial associations have been shown. These studies have spurred research into a number of complex diseases with unclear etiologies where a connection to the microbiome is suspected.

Overall, our current understanding of the precise relationships between the human gut microbiome and disease remains limited. Existing case–control studies often report finding disease-associated microbial "dysbiosis". However, the term "dysbiosis" is inconsistently and often vaguely defined, and can have a wide range of interpretations[10, 11]. Thus, we lack a comprehensive understanding of precisely how microbial communities and specific microbes within those communities cause, respond to, or contribute to disease. Are different diseases characterized by distinct shifts in the gut microbiome? Are some diseases marked by an invasion of pathogens, whereas others show a depletion of beneficial bacteria? Can we identify microbial biomarkers for certain conditions, which are consistently enriched or depleted in a disease across many patient cohorts? Finally, are some bacteria part of a non-specific "healthy" or "diseased" microbiome and consistently associated with health or disease in general?

One approach to synthesize existing knowledge is to identify consistencies across studies through a meta-analysis, which allows researchers to find and remove false positives and negatives that may obscure underlying biological patterns. However, prior meta-analyses of case–control gut microbiome studies have yielded mixed results and did not contextualize their findings across multiple diseases[12–14]. For some conditions like inflammatory bowel disease (IBD), an overall difference in the gut microbiota was found within several studies, but no individual microbes were consistently associated with IBD across studies[12]. For other conditions like obesity, multiple meta-analyses have found little to no difference in the gut microbiomes of obese and lean patients[12–14], even though the microbiome has been causally linked to obesity in mouse models[3, 15]. These meta-analyses have been limited by focusing on only one or two diseases, and thus do not extend their findings across a broader landscape of human disease to answer more general questions about overall patterns of disease-associated microbiome shifts.

In this paper, we collected 28 published case–control 16S amplicon sequencing gut microbiome data sets spanning ten different disease states. We acquired raw data and disease meta-data for each study and systematically re-processed and re-analyzed the data. We investigated whether consistent and specific disease-associated changes in gut microbial communities could be identified across multiple studies of the same disease. Certain diseases (e.g., colorectal cancer (CRC)) are marked by an enrichment of disease-associated bacteria, while others (e.g., IBD) are characterized by a depletion of health-associated bacteria. Some conditions (e.g., diarrhea) exhibit large-scale community shifts with many associated microbes, while most show only a handful of associations. However, many associations are not specific to individual diseases but rather respond to multiple disease states. In most studies, the majority of the individual disease-associated microbes were part of this set of bacteria that respond non-specifically to healthy and diseased states. Thus, associations from individual case–control studies should be interpreted with caution, as these microbes may be indicative of a shared response to disease rather than part of disease-specific differences. Together, these findings reveal distinct categories of dysbioses, which can inform the development of microbiome-based diagnostics and therapeutics.

## Results

**Most disease states show altered microbiomes.** To answer questions about the reproducibility and generalizability of reported associations between the human microbiome and disease, we collected, re-processed, and re-analyzed raw data from a collection of microbiome data sets. We included studies with publicly available 16S amplicon sequencing data (i.e., FASTQ or FASTA) for stool samples from at least 15 case patients, which also had associated disease metadata (i.e., case or control disease labels). Studies which exclusively focused on children under 5 years old were excluded from our analyses. We identified over 50 suitable case–control 16S data sets, of which 28 were successfully downloaded, processed, and included in a publicly available database, which we called MicrobiomeHD[16]. Characteristics of these data sets, including sample sizes, diseases and conditions, and references, are shown in Table 1 and Supplementary Table 1. For each downloaded study, we processed the raw sequencing data through our 16S processing pipeline (https://github.com/thomasgurry/amplicon_sequencing_pipeline) (see Supplementary Tables 2 and 3 for detailed data sources and processing methods). 100% de novo OTUs were assigned taxonomy with the RDP classifier[17] ($c = 0.5$), converted to relative abundances by dividing by total sample reads, and collapsed to the genus level. OTUs which were not assigned at the genus level were discarded. By collapsing data to the genus level, we lost the sensitivity to detect fine-scale differences in species or strain abundances across case and control groups, but we minimized certain batch effects that plague comparisons across studies. Thus, we took a course-grained approach to optimize our ability to compare data across studies at the expense of phylogenetic resolution.

We first asked whether reported associations between the gut microbiome and disease would be recapitulated once we controlled for processing and analysis approaches. To test whether the gut microbiome is altered in a variety of disease states, we built genus-level random forest classifiers to classify cases from controls within each study. We compared the resulting area under the receiver operating characteristic (ROC) curves (AUC) across studies (Fig. 1a and Supplementary Fig. 1). We could classify cases from controls (AUC > 0.7) for at least one data set for all diseases except arthritis and Parkinson's disease, which each only had one study. Notably, all diarrhea data sets (except Youngster et al.[18], which had only four distinct control patients and thus was not included in this analysis) had very high classifiability (AUC > 0.9). We successfully classified patients from controls (AUC > 0.7) in three out of four IBD studies and all four CRC studies, which is consistent with previous work showing that these patients can be readily distinguished from controls using supervised classification methods[12, 19–21]. Thus, the microbiome is indeed altered in many different diseases.

**Loss of beneficial microbes or enrichment of pathogens**. We next wondered whether the specific type of alteration was consistent across independent cohorts of patients with the same disease. We performed univariate tests on genus-level relative abundances for each data set independently and compared results across studies (Kruskal–Wallis (KW) test with the Benjamini–Hochberg false discovery rate (FDR) correction[22]). Our re-analyses of the studies were largely consistent with the originally reported results. The same taxonomic groups showed similar trends as in the original publications, despite differences in data-processing methodologies (see Supplementary Note 1 for a full comparison of our re-analysis with previously published results). Furthermore, we found that the disease-associated changes in the microbiome could be categorized into meaningful groups, which provide insight into possible etiologies or therapeutic strategies for different types of disease.

In some diseases, microbiome shifts are dominated by an enrichment of a small number of "pathogenic" bacteria. In these cases, it is possible that the microbes play a causal role and that they could be targeted with narrow-spectrum anti-microbials. Colorectal cancer is characterized by such a shift, and we found significant agreement across three of the four CRC studies[8, 20, 21, 23] (Figs. 1b and 2, genus labels in Supplementary Fig. 2). Dysbiosis associated with CRC is generally characterized by increased prevalence of the known pathogenic or pathogen-associated *Fusobacterium*, *Porphyromonas*, *Peptostreptococcus*, *Parvimonas*, and *Enterobacter* genera (i.e., these genera were higher in CRC patients in two or more studies, Figs. 2 and 3a, genus labels in Supplementary Figs. 2 and 3). *Fusobacterium* is associated with a broad spectrum of human diseases and *Porphyromonas* is a known oral pathogen[24, 25].

By contrast, other disease-associated microbiome shifts are characterized by a depletion of health-associated bacteria in patients relative to controls. In these cases, probiotics that replace missing taxa may be a better treatment strategy than anti-microbials. Across our four IBD studies, patient microbiomes were dominated by a depletion of genera in patients relative to controls, especially butyrate-producing *Clostridiales*[19, 26–28] (Figs. 1b and 2, genus labels in Supplementary Fig. 2). In particular, five genera from the *Ruminococcaceae* and *Lachnospiracaea* families were consistently depleted in IBD patients relative to controls in at least two studies (Fig. 3a, genus labels in Supplementary Fig. 3). While not all genera within *Ruminococcaceae* and *Lachnospiracaea* are verified short chain fatty acid (SCFA) producers, the dominant genera within these families are known to harbor genes for short chain fatty acid production[29] and are often associated with colonic health[30–32]. We found similar results when comparing Crohn's disease and ulcerative colitis patients to controls separately, without any consistent patterns across data sets that distinguished either IBD subtype (Supplementary Note 2; Supplementary Figs. 4 and 5).

Some conditions are characterized by a broad restructuring of gut microbial communities. In these cases, full community restoration strategies like fecal microbiota transplants may be more appropriate. For example, diarrhea consistently results in large-scale rearrangements in the composition of the gut microbiome, which is likely reflective of reduced stool transit time (Figs. 1 and 2). We saw many microbes consistently

**Table 1 Data sets collected and processed through standardized pipeline**

| Dataset ID | Controls | N (controls) | Cases | N (cases) | Reference |
|---|---|---|---|---|---|
| Singh 2015, EDD | H | 82 | EDD | 201 | 35 |
| Schubert 2014, CDI | H | 154 | CDI | 93 | 33 |
| Schubert 2014, non-CDI | H | 154 | non-CDI | 89 | 33 |
| Vincent 2013, CDI | H | 25 | CDI | 25 | 34 |
| Youngster 2014, CDI | H | 4 | CDI | 19 | 18 |
| Goodrich 2014, OB | H | 428 | OB | 185 | 43 |
| Turnbaugh 2009, OB | H | 61 | OB | 195 | 42 |
| Zupancic 2012, OB | H | 96 | OB | 101 | 44 |
| Ross 2015, OB | H | 26 | OB | 37 | 45 |
| Zhu 2013, OB | H | 16 | OB | 25 | 1 |
| Baxter 2016, CRC | H | 172 | CRC | 120 | 20 |
| Zeller 2014, CRC | H | 75 | CRC | 41 | 21 |
| Wang 2012, CRC | H | 54 | CRC | 44 | 8 |
| Chen 2012, CRC | H | 22 | CRC | 21 | 23 |
| Gevers 2014, IBD | non-IBD | 16 | CD | 146 | 26 |
| Morgan 2012, IBD | H | 18 | UC, CD | 108 | 27 |
| Papa 2012, IBD | non-IBD | 24 | UC, CD | 66 | 19 |
| Willing 2010, IBD | H | 35 | UC, CD | 45 | 28 |
| Noguera-Julian 2016, HIV | H | 34 | HIV | 205 | 39 |
| Dinh 2015, HIV | H | 15 | HIV | 21 | 41 |
| Lozupone 2013, HIV | H | 13 | HIV | 23 | 40 |
| Son 2015, ASD | H | 44 | ASD | 59 | 7 |
| Kang 2013, ASD | H | 20 | ASD | 19 | 2 |
| Alkanani 2015, T1D | H | 55 | T1D | 57 | 59 |
| Mejia-Leon 2014, T1D | H | 8 | T1D | 21 | 60 |
| Wong 2013, NASH | H | 22 | NASH | 16 | 61 |
| Zhu 2013, NASH | H | 16 | NASH | 22 | 1 |
| Scher 2013, ART | H | 28 | PSA, RA | 86 | 52 |
| Zhang 2013, LIV | H | 25 | CIRR, MHE | 46 | 51 |
| Scheperjans 2015, PAR | H | 74 | PAR | 74 | 9 |

Non-CDI controls are patients with diarrhea who tested negative for *C. difficile* infection. Non-IBD controls are patients with gastrointestinal symptoms but no intestinal inflammation. Data sets are ordered as in Fig. 1
*ART* arthritis, *ASD* autism spectrum disorder, *CD* Crohn's disease, *CDI* *Clostridium difficile* infection, *CIRR* liver cirrhosis, *CRC* colorectal cancer, *EDD* enteric diarrheal disease, *H* healthy, *HIV* human immunodeficiency virus, *LIV* liver diseases, *MHE* minimal hepatic encephalopathy, *NASH* non-alcoholic steatohepatitis, *OB* obesity, *PAR* Parkinson's disease, *PSA* psoriatic arthritis, *RA* rheumatoid arthritis, *T1D* type I diabetes, *UC* ulcerative colitis

associated with both *Clostridium difficile* infection (CDI) and non-CDI diarrhea (Figs. 2 and 3a)[18], [33–35]. In general, Proteobacteria increase in prevalence in patients with diarrhea, with a concomitant decrease in the relative abundances of Bacteroidetes and some Firmicutes. In particular, we see a reduction in butyrate-producing Clostridia, including genera within *Ruminococcaceae* and *Lachnospiraceae* families, which have been associated with a healthy gut[36]. We also see an increase in prevalence of genera that contain organisms often associated with lower pH and higher oxygen levels of the upper gut, like *Lactobacillaceae* and *Enterobacteriaceae*, in patients with diarrhea (Fig. 3a)[37]. Additionally, both CDI and non-CDI diarrhea patients had lower alpha diversity, a measure of overall community structure, than healthy controls in all studies (Supplementary Figs. 6–8). Consistent with the CDI and non-CDI diarrheal studies, we also found that organisms associated

with the upper gut, like *Lactobacillus* and *Enterobacteriaceae*, appear to be enriched in IBD patients, who can present with diarrheal symptoms (Supplementary Fig. 2)[37], [38]. IBD patients also tended to have lower alpha diversities than controls (Crohn's disease vs. controls in three studies, ulcerative colitis vs. controls in two studies; Supplementary Figs. 6–8), though this difference was less drastic than in the diarrheal studies where all patients had active diarrhea.

In some studies, confounding variables may drive associations. For example, there were no consistent differences between cases and controls across HIV studies because of demonstrated confounders[39–41] (Figs. 2 and 3a). As in the original Lozupone et al.[40] study, we found enrichment in *Prevotella*, *Catenibacterium*, *Dialister*, and *Desulfovibrio* in HIV-positive patients, in addition to eight other genera (Fig. 2 and Supplementary Fig. 2). We also found depletion of *Bacteroides*,

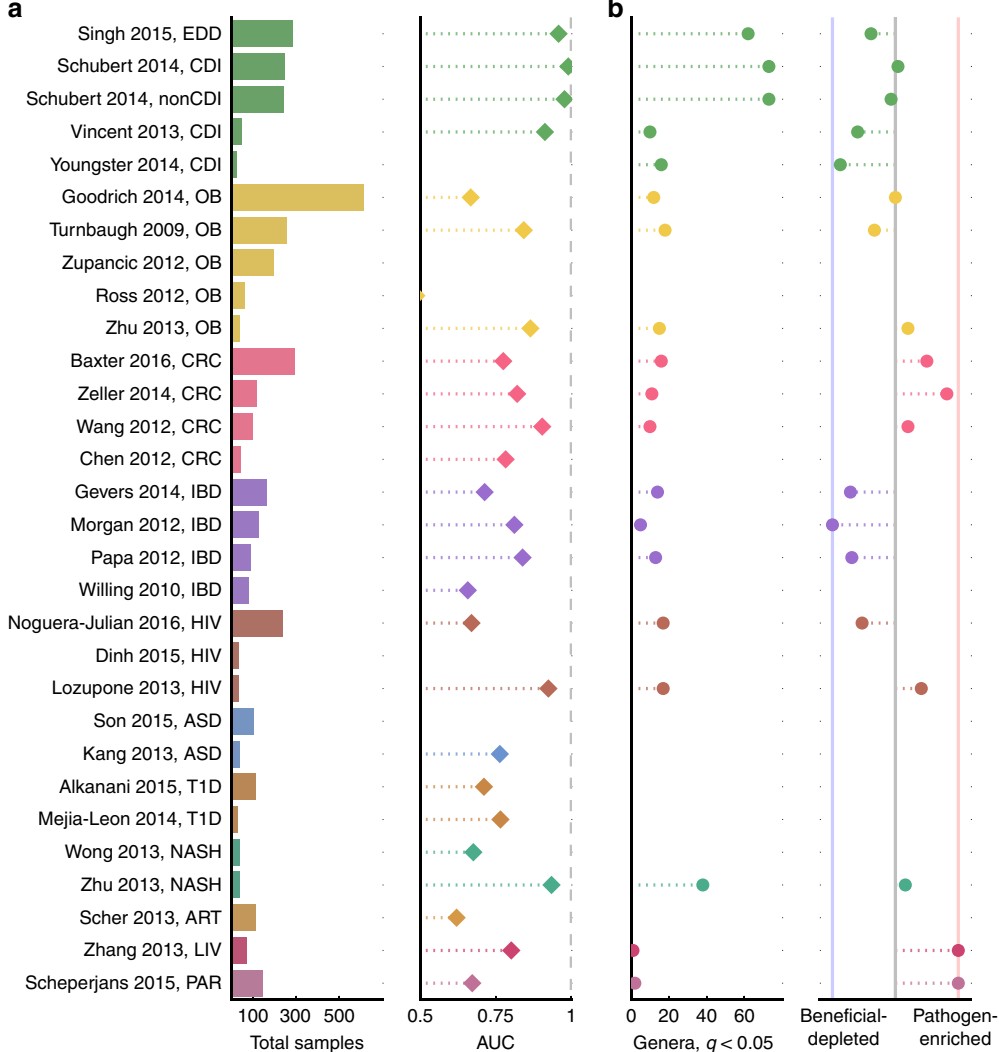

**Fig. 1** Most diseases show microbiome alterations, and consistent disease-associated shifts differ in their extent and direction. **a** Left: Total sample size for each study included in these analyses. Additional information about each data set can be found in Table 1. Studies on the y-axis are grouped by disease and ordered by decreasing sample size (top to bottom). Right: Area under the ROC curve (AUC) for genus-level random forest classifiers. *X*-axis starts at 0.5, the expected value for a classifier which assigns labels randomly, and AUCs less than 0.5 are not shown. ROC curves for all data sets are in Supplementary Fig. 1. Note that Youngster et al.[18] had only four distinct control patients was excluded from the random forest analysis. **b** Left: Number of genera with $q < 0.05$ (Kruskal–Wallis (KW) test, Benjamini–Hochberg FDR correction) for each data set. If a study has no significant associations, no point is shown. Right: Direction of the microbiome shift, i.e., the percent of total associated genera which were enriched in diseased patients. In data sets on the leftmost blue line, 100% of associated ($q < 0.05$, FDR KW test) genera are health-associated (i.e., depleted in patients relative to controls). In data sets on the rightmost red line, 100% of associated ($q < 0.05$, FDR KW test) genera are disease-associated (i.e., enriched in patients relative to controls). Supplementary Figs. 14 and 15 show q values and effects for each genus in each study

*Odoribacter*, *Anaerostipes*, *Parasutterella*, and *Alistipes* in HIV-positive patients relative to controls. However, the Noguera-Julian et al.[39] study showed that the genera that were significantly associated with HIV in the Lozupone paper were strongly associated with sexual behavior (e.g., men who have sex with men were associated with much higher *Prevotella* levels), and our re-analysis also found conflicting results between these two studies (Fig. 2). Thus, there is no consensus on what genera are associated with HIV. Obesity is another example where

confounding variables may drive microbiome alterations. Three recent meta-analyses found no reproducible obesity-associated microbiome shifts[12–14], which is consistent with our classification results where we were only able to accurately classify obese and control patients in two out of five studies (Zhu et al.[1], Turnbaugh et al.[42]; Fig. 1a). Our genus-level re-analysis did find a few consistent genus-level associations between lean and obese patients[1, 42–45]. Two genera, *Roseburia* and *Mogibacterium*, were significantly enriched in obese individuals across two of the

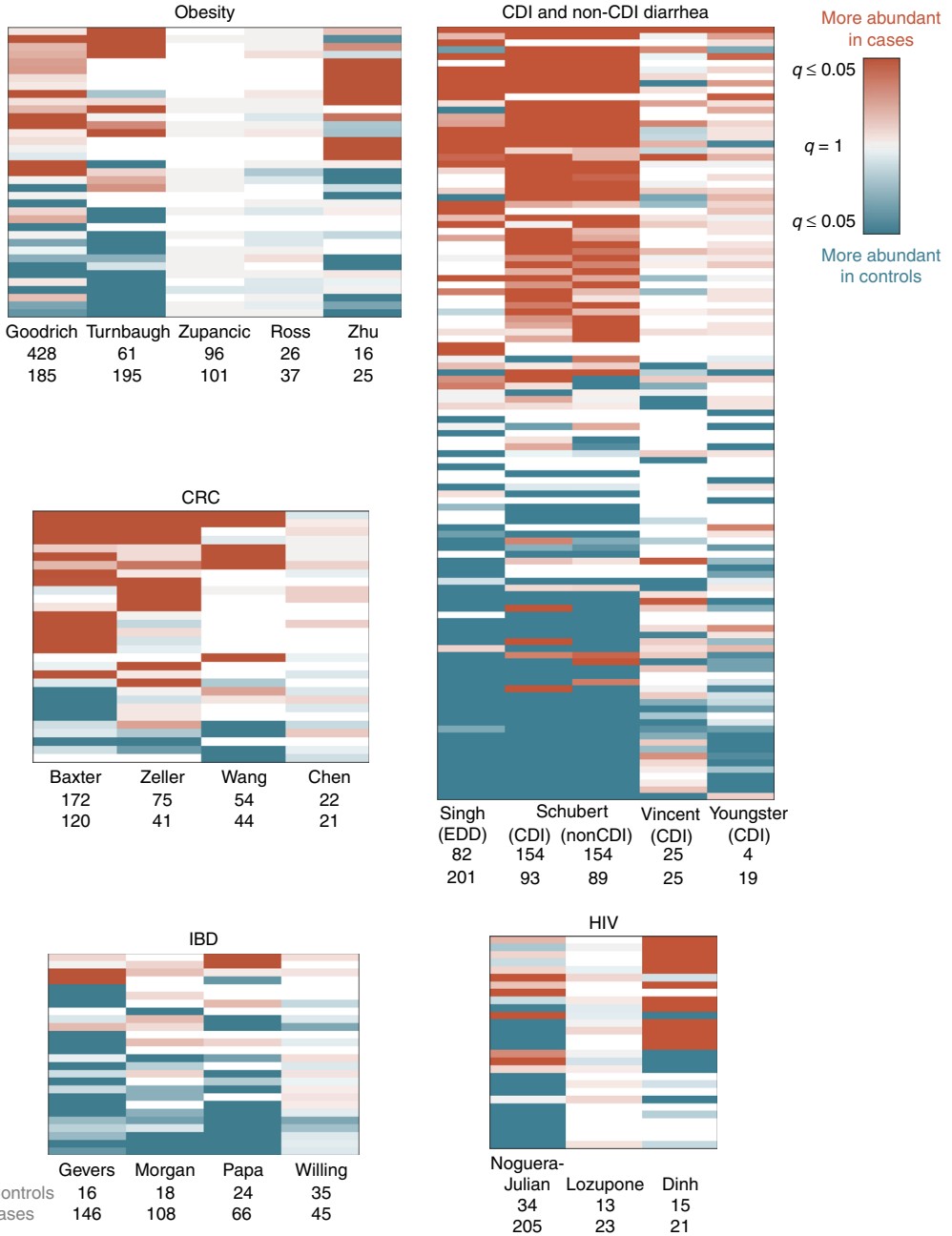

**Fig. 2** Comparing results from multiple studies of the same disease reveals patterns in disease-associated microbiome alterations. Heat maps showing $\log_{10}(q$ values) for each disease (KW test, Benjamini–Hochberg FDR correction). Rows include all genera which were significant in at least one data set within each disease, columns are data sets. $q$ values are colored by direction of the effect, where red indicates higher mean abundance in disease patients and blue indicates higher mean abundance in controls. Opacity ranges from $q = 0.05$–1, where $q$ values less than 0.05 are the most opaque and $q$ values close to 1 are gray. White indicates that the genus was not present in that data set. Within each heat map, rows are ordered from most disease-associated (top) to most health-associated (bottom) (i.e., by the sum across rows of the $\log_{10}(q$ values), signed according to directionality of the effect). The extent of a disease-associated microbiome shift can be visualized by the number of rows in each disease heat map; the directionality of a shift can be seen in the ratio of red rows to blue rows within each disease. See Supplementary Fig. 2 for genus (row) labels

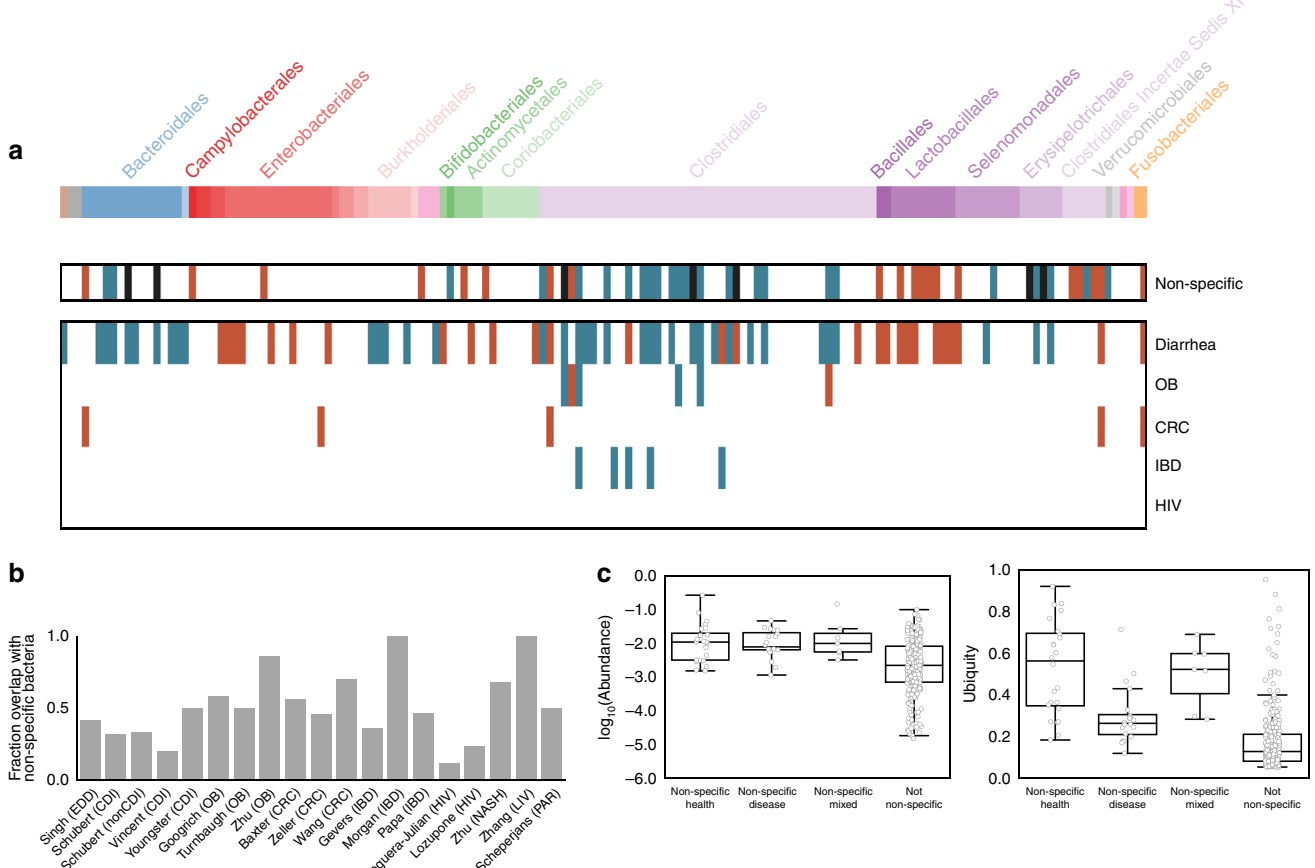

**Fig. 3** The majority of disease-associated microbiome associations overlap with a non-specific microbial response to disease. **a** Non-specific and disease-associated genera. Genera are in columns, arranged phylogenetically according to a PhyloT tree built from genus-level NCBI IDs (http://phylot.biobyte.de). Non-specific genera are associated with health (or disease) in at least two different diseases (q < 0.05, KW test, Benjamini–Hochberg FDR correction). Disease-specific genera are significant in the same direction in at least two studies of the same disease (q < 0.05, FDR KW test). As in Fig. 2, blue indicates higher mean abundance in controls and red indicates higher mean abundance in patients. Black bars indicate mixed genera which were associated with health in two diseases and also associated with disease in two diseases. Disease-specific genera are shown for diseases with at least three studies. Phyla, left to right: Euryarchaeota (brown), Verrucomicrobia Subdivision 5 (gray), Candidatus Saccharibacteria (gray), Bacteroidetes (blue), Proteobacteria (red), Synergistetes (pink), Actinobacteria (green), Firmicutes (purple), Verrucomicrobia (gray), Lentisphaerae (pink), Fusobacteria (orange). See Supplementary Fig. 3 for genus labels. **b** The percent of each study's genus-level associations which overlap with the shared response (q < 0.05, FDR KW test). Only data sets with at least one significant association are shown. **c** Overall, abundance and ubiquity of non-specific genera across all patients in all data sets. Non-specific genera on the x-axis are as defined above

obesity studies (Fig. 3a). Furthermore, *Anaerovorax*, *Oscillibacter*, *Pseudoflavonifractor*, and *Clostridium IV* were depleted in obese patients relative to controls in two of the studies. However, two of the five studies had no significant genus-level associations (q < 0.05), despite one having a large sample size (Zupancic et al.[44]). This suggests that confounding factors like diet may have given rise to certain associations found in our re-analysis and previously reported in the literature[14]. More studies that control for potential confounders, like host behavior and diet, will be required for diseases like obesity and HIV, where associations with the microbiome remain unclear. Finally, patients in case–control cohorts are frequently on other medications such as antibiotics which may confound disease-associated micro-biome shifts. Six of our data sets included antibiotics metadata, and of these only one data set (Schubert et al.[33]) had more than five controls who were on antibiotics. Thus, it is very likely that disease-associated genera in conditions which are often treated with antibiotics (e.g., diarrhea, IBD) are confounded with antibiotic usage. Future case–control studies should focus on better separating treatment and disease variables by collecting detailed metadata on antibiotic and other medication usage, and

perhaps also by recruiting controls undergoing a variety of treatments.

**Shared vs. disease-specific microbial responses**. Finally, we sought to determine whether a unified microbiome response to general health and disease could be identified. Previous studies have proposed that reduced alpha diversity is a reliable indicator of disease-associated dysbiosis[34, 42, 46]. In our re-analysis, we found no consistent reduction of alpha diversity in case patients, with the exception of diarrhea and perhaps IBD (Supplementary Figs. 6–8). These results are consistent with previous meta-analyses, which found inconsistent relationships between alpha diversity and disease and very small effect sizes in non-diarrheal diseases[12, 13]. To further address the question of whether we could find a robust, generalized signal for diseased microbiomes regardless of the disease type, we built random forest classifiers to distinguish healthy patients from any type of case patient. The AUCs from these general healthy vs. disease classifiers correlated strongly with the original single data set classification results, indicating that there is indeed a general microbiome signal that

can be identified even across different diseases (see Supplementary Note 3 and Supplementary Fig. 9).

Having putatively shown the presence of a generalized microbial response to disease, we next sought to identify individual genera which respond non-specifically to health and disease. We considered a genus to be part of the non-specific, shared microbial response if it was significantly enriched or depleted ($q < 0.05$) in at least one data set from at least two different diseases (see Supplementary Note 4 and Supplementary Figs. 10 and 11 for further discussion on alternative definitions and statistical significance of shared response). We identified 24 health-associated genera and 20 disease-associated genera out of the 152 genera that were significant in at least one data set (Fig. 3a, genus labels in Supplementary Fig. 3). We also found seven genera that were both health- and disease-associated (i.e., they were enriched in controls across at least two diseases, but were also depleted in controls in different comparisons across at least two diseases) (Fig. 3a, black). Perhaps these genera represent bacteria disproportionately affected by confounders or technical artifacts. Alternatively, different species or strains within these genera may play alternate roles across diseases or community contexts, giving rise to variable responses at the genus level.

We identified distinct subgroups of microbes within the *Bacteroidetes* and *Firmicutes* phyla that respond non-specifically to health and disease (Fig. 3a). The order *Clostridiales* (specifically the *Lachnospiraceae* and *Ruminococcaceae* families) is associated with health across multiple diseases while the order *Lactobacillales* and family *Clostridiales Incertae Sedis XI* are associated with disease. The majority of the the non-specific responders in the order *Clostridiales* were associated with health, comprising the majority of all of the microbes which were non-specifically associated with healthy patients (17 genera out of 24 total health-associated genera). All five of the non-specific responders in the order *Lactobacillales* were enriched in case patients across multiple diseases. *Lactobacillales* genera are adapted to the lower pH of the upper gastrointestinal tract[37]. Perhaps the shared disease-associated taxa are indicators of shorter stool transit times and disruptions in the redox state and/or pH of the lower intestine, rather than specific pathogens. These non-specific responders are consistent with the results from a recent meta-analysis of six metagenomics data sets, which also found *Lactobacillales* and *Clostridiales* microbes among the most discriminative classification features across multiple studies[47]. Finally, we found that the order *Bacteroidales* is more mixed: two *Bacteroidales* genera were non-specifically associated with health, one with disease, and two with both health and disease.

A majority of bacterial associations within individual studies overlap with the shared response. For each data set that had at least one significant ($q < 0.05$) association, we calculated the percent of associated genera which were also part of the non-specific response in the same direction (Fig. 3b). Strikingly, the majority of microbial responses were not specific to individual diseases; on average, 51% of a data set's genus-level associations were genera that were associated with more than one disease. In light of this finding, it is important that researchers performing future case–control studies consider whether an identified microbial association is truly specific to their disease of interest or is instead responding to a common symptom (e.g., diarrhea) or perhaps generally associated with health or sickness. Additionally, they can use the knowledge that many microbes respond non-specifically to disease to narrow putative causal or diagnostic biomarkers to microbes which fall outside of the shared response, and are thus more likely to be specific to the disease being studied. Researchers can access an updated list of shared microbial responders from this analysis at the MicrobiomeHD

database[16], or they can curate their own lists by performing similar cross-disease meta-analyses.

Bacteria which are non-specifically associated with health are both ubiquitous and abundant across people, whereas bacteria which are non-specifically associated with disease are abundant when present but are not ubiquitous. We calculated the average relative abundance (i.e., the total relative abundance across all patients divided by the number of patients with non-zero abundance) and ubiquity (i.e., the number of patients with non-zero abundance divided by the total number of patients) for each genus in the shared response. We found that health-associated genera were more ubiquitous than disease-associated ones, but not necessarily more abundant (Fig. 3c). Thus, presence/absence of the non-specifically disease-associated genera appears to be a better indicator of disease-associated microbial shifts than changes in their relative abundances. However, a small subset of the non-specifically disease-associated genera were relatively ubiquitous across patients. Among the most ubiquitous were *Escherichia/Shigella* and *Streptococcus*. *Escherichia* includes common commensal strains as well as pathogenic strains[48], and is frequently present in healthy people's guts as well as over-represented in sick patients. Genera within *Enterobacteriaceae*, *Lactobacillaceae*, and *Streptococcaceae* families are dominant in the upper gastrointestinal tract[37, 49] and are present in many people's stool at low frequency. These taxa likely become enriched with faster stool transit time (i.e., signatures of diarrhea)[37, 50].

**Within and cross-disease meta-analysis improves interpretability**. Identifying disease-specific and non-specific microbial responses required comparing studies both within and across multiple diseases. Multiple studies of the same disease were necessary to identify shifts consistently associated with individual diseases. We did not find consistent bacterial associations for conditions with fewer than four data sets (Figs. 1 and 3a). Within-disease meta-analysis also increased our ability to interpret the results from any one data set. Despite few significant differences, some of these studies (e.g., Zhang et al.[51], Zhu et al.[1]) had high classifiability of patients vs. controls (AUC > 0.7, Fig. 1a), indicating that there may be a disease-associated shift that was not detected by univariate comparisons. However, because few other studies of the same disease were available for comparison, we could not confidently interpret the classification results beyond the reported AUC. For other studies with high AUCs but few univariate associations (e.g., Vincent et al.[34], Morgan et al.[27], and Chen et al.[23]), our confidence that the high AUCs reflect true disease-associated differences increased because the high AUCs were consistent with other classifiers from the same disease type.

Meta-analysis identified potential false positives and false negatives across studies and conditions. For example, we found that reported associations between alpha diversity and disease within individual studies tended to lose significance when looking across studies, except in the case of diarrhea and perhaps IBD (Supplementary Figs. 6–8). Another example of a potential false positive was the association between *Prevotella* and disease. Autism[2], rheumatoid arthritis[52], and HIV[40, 41] have each been reported to be associated with *Prevotella*. For each of these diseases, the associations with *Prevotella* were weakly significant or complicated by confounding factors. In our more statistically conservative re-analysis, we found no association between autism or arthritis and *Prevotella*. As mentioned previously, in the case of HIV, the association with *Prevotella* was due to demographic factors unrelated to disease[39]. Regardless of whether shifts in *Prevotella* are truly biologically related to each studied disease state, it is clear that such shifts are not specific to one particular condition and should not be reported as putative disease-specific

biomarkers. We also found that certain signals picked out by meta-analysis did not always hold within individual studies. For example, studies with small sample sizes often had few or no significant associations (e.g., Vincent et al.[34], Chen et al.[23], and Willing et al.[28]). Here the fact that other studies analyzing the same diseases consistently found associations strengthens the hypothesis that the lack of microbiome-associated signal in these studies was due to low power rather than a lack of true signal. Because individual studies are plagued by low statistical power, confounding variables, and batch effects which can obscure biological signals, the identification of disease-specific and non-specific microbial associations will continue to improve as more data sets and diseases are included in future meta-analyses.

## Discussion

Here we report patterns of disease-associated shifts in the human gut microbiome that differ in their directionality (i.e., fraction of disease-enriched vs. disease-depleted genera) and extent (i.e., total number of genera that differ between cases and controls). Some diseases are characterized by an invasion of pathogenic or disease-associated bacteria (e.g., CRC), while others largely show a depletion of health-associated microbes (e.g., IBD). Diarrheal illnesses induce large-scale rearrangement of many members of the microbiota, whereas other conditions show fewer associations. We also find a set of microbes which are non-specifically associated with multiple diseases and show that these microbes comprise many of the disease-associated genera within any given study.

The identification of a non-specific microbial response is an important concept that should be considered in future case–control microbiome studies. It suggests that studies should be interpreted with extra caution, as many identified microbial associations may be indicative of a shared response to health or disease rather than a disease-specific biological difference. Microbes that are non-specifically associated with multiple diseases would not be useful as disease-specific diagnostics or to address causality[10]. On the other hand, bacteria that are associated with healthy patients across multiple diseases could be developed into a general probiotic which may be suited for many different conditions.

Additionally, characterizing "dysbioses" by their directionality and extent is a useful framework to generate hypotheses for future research on complex, heterogenous diseases with links to the microbiome. For example, the search for microbiome-based diagnostics may be more appropriate for diseases with consistently enriched disease-associated microbes, like CRC. On the other hand, patients with diseases which are characterized by depletion of health-associated microbes, like IBD, may benefit from prebiotic or probiotic interventions designed to enrich for these taxa. Furthermore, conditions which are characterized by large-scale shifts in community structure may be well suited to treatment with fecal microbiota transplantation, as in CDI[18]. While many of these conditions are unlikely to be fully treated by antibiotics, probiotics, or fecal microbiota transplants, our proposed framework could guide the search for new therapies and etiologies by generating testable hypotheses with higher likelihoods of success[10].

This analysis is the first to compare microbiome studies across more than two different diseases and highlights the importance of making raw data and associated patient metadata publicly available to enable future, more comprehensive analyses. This analysis does not include all possible studies, and certain important gastrointestinal diseases (e.g., irritable bowel syndrome) are missing, largely due to data and metadata availability. Future studies should expand on this work by including more cohorts from the same diseases as well as more diseases. To re-analyze these studies, we applied standard methods commonly used in the field and assumed that the original study designs and patient selection methods were adequate. We were reassured to find that a straightforward and standardized approach was able to recover very similar results to those previously reported in the various papers. Thus, we did not formally investigate heterogeneity between cohorts or technical inter-study batch effects. However, it is clear from our genus-level results that there is significant variation even across studies of the same disease. There are many possible reasons for this variation (experimental and sequencing artifacts, host-related covariates, stochastic disease-associated community changes, etc.[11, 53, 54]), and future analyses should consider methods to correct for host confounders and technical batch effects. Concerns about batch effects motivated us to analyze the data at the genus level, which necessarily limited our resolution and biological interpretations of identified associations (e.g., different species or strains within a genus may have different associations with disease, which would not be captured in this analysis). Making raw data from case–control studies publicly available will also allow researchers to develop methods to correct for these batch effects, in addition to enabling more comprehensive future meta-analyses.

Despite the limitations of this study, our results provide more nuanced insight into dysbiosis, revealing distinct types of alterations that more precisely describe disease-associated microbiome shifts. As the number of case–control cohorts increases, similar meta-analyses could be used to compare related diseases and identify microbiome alterations associated with general host physiological changes. For example, there may be a group of microbes which respond to or cause systemic inflammation. Could we identify these microbes by comparing multiple inflammatory or auto-immune diseases and study them to better understand the interactions between the microbiome and our immune system? Furthermore, some microbes may be consistently associated with neurological conditions and could contribute to the gastrointestinal symptoms that accompany or precede neurological manifestations[2, 9]. Studying these microbes could help us understand the "gut-brain axis" by identifying common neuroactive molecules produced by these bacteria, which could also be used as targets for new treatments[4–6]. Finally, meta-analysis could be used to identify subsets of patients who exhibit distinct microbiome shifts within heterogenous diseases like IBD or in conditions which exhibit stochastic microbial responses, allowing for further stratification of disease subtypes and microbiome disruptions[11, 28, 55]. This work demonstrates that employing standard methods to contextualize new results within the broader landscape of clinically relevant microbiome studies is feasible and adds value to individual analyses. As excitement in this field grows, researchers should harness the increasing number of replicated case–control studies to swiftly and productively advance microbiome science from putative associations to transformative clinical impact.

## Methods

**Data set collection**. We identified case–control 16S studies from keyword searches in PubMed and by following references in meta-analyses and related case–control studies. We included studies with publicly available raw 16S data (fastq or fasta) and metadata indicating case or control status for each sample. Most data were downloaded from online repositories (e.g., SRA) or links provided in the original publications, but some were acquired after personal communication with the authors (Supplementary Table 3). We did not include any studies which required additional ethics committee approvals or authorizations for access (e.g., controlled dbGaP studies). In studies where multiple body sites were sampled or where multiple samples were taken per patient, we also required the respective metadata to include those metadata. We analyzed only stool 16S samples, and excluded studies with fewer than 15 case patients. In CRC studies with multiple control groups (e.g., healthy and non-CRC adenoma), only the healthy patients were used

as controls for all of our comparisons. In studies with non-healthy controls (e.g., non-IBD patients), these patients were used as controls (as in the original papers). In the Schubert et al. CDI study[33], which had both CDI and non-CDI diarrheal patients, each group was used as an independent case group compared with controls. We also analyzed the NASH and obese patients from the Zhu et al. study[1] as independent case groups. When obesity studies reported body mass index instead of obesity status, we considered patients with BMI less than 25 as our control group and patients with BMI greater than 30 as the case group.

**16S processing**. Raw data were downloaded and processed through our in-house 16S processing pipeline (https://github.com/thomasgurry/amplicon_sequencing_pipeline). Data and metadata were acquired as described in Supplementary Table 3. When needed, we de-multiplexed sequences by finding exact matches to the provided barcodes and trimmed primers with a maximum of one mismatch. In general, sequences were quality filtered by truncating at the first base with quality score $Q < 25$. However, some data sets did not pass this stringent quality threshold (i.e., the resulting OTU table was either missing many of the original samples, or the read depth was significantly lower than reported in the original paper). For 454 data, we loosened the quality threshold to 20, whereas for paired-end Illumina data, we removed reads with more than two expected errors. If possible, all reads were trimmed to 200 bp. In cases where this length trimming discarded a majority of sequences, we lowered our threshold to 150 or 101 bp. The specific processing parameters we used for each data set can be found in Supplementary Table 2. To assign OTUs, we clustered OTUs at 100% similarity using USEARCH[56] and assigned taxonomy to the resulting OTUs with the RDP classifier[17] and a confidence cutoff of 0.5. For each data set, we removed samples with fewer than 100 reads and OTUs with fewer than 10 reads, as well as OTUs which were present in fewer than 1% of samples within a study. We calculated the relative abundance of each OTU by dividing its value by the total reads per sample. We then collapsed OTUs to genus level by summing their respective relative abundances, discarding any OTUs which were un-annotated at the genus level. All statistical analyses were performed on this genus-level relative abundance data.

**Statistical analyses**. To perform supervised classification of cases and controls within each data set, we built Random Forest classifiers with fivefold cross-validation. To build our train and test sets, we used the python scikit-learn StratifiedKFold function with shuffling of the data[57]. To build our classifiers, we used the RandomForestClassifier function with 1000 estimators and other default settings[57]. We found no significant effect of various Random Forest parameters on the AUCs (Supplementary Figs. 12 and 13). We calculated the interpolated area under the ROC curve (AUC) for each classifier based on the cross-validation testing results. To account for spurious high classifiability due to class imbalances, we also calculated the Cohen's kappa score for each classifier using sklearn.metrics. cohen_kappa_score on the test set predictions (Supplementary Table 4). The kappa scores correlated well with the AUCs (Pearson $\rho = 0.9$), indicating that the majority of the classifiers performed well even when considering their underlying data distributions. We excluded Youngster et al.[18], which had only four distinct control patients, from all classifier analyses.

We performed univariate analyses on the relative abundances of genera in cases and controls with a non-parametric Kruskal–Wallis test using the scipy.stats. mstats.kruskalwallis function[58]. We corrected for multiple hypothesis testing in each data set with the Benjamini–Hochberg false discovery rate using statsmodels. sandbox.stats.multicomp.multipletests with method='fdr_bh'[22]. We performed all univariate analyses on genus-level relative abundances within each dataset individually, and then compared these results across all studies.

We considered a genus to be consistently associated with a disease (Fig. 3a, bottom) if it was significantly associated ($q < 0.05$) with the disease in the same direction in at least two studies of that disease. We considered a genus to be a non-specific microbial association (Fig. 3a, top) if it was significantly associated ($q < 0.05$) in at least one data set of at least two different diseases in the same direction. When we defined these non-specific genera, we did not include data sets which used non-healthy controls (Papa et al.[19] and Gevers et al.[26]) and the Lozupone et al. data set[40], where the microbiome signal reflected behavior rather than disease state[39].

To build our generalized healthy vs. disease classifiers (Supplementary Fig. 9), we first concatenated metadata and genus-level abundance data for all data sets that had healthy controls (i.e., all data sets except Papa et al.[19] and Gevers et al.[26], which used non-IBD patients as controls, and CDI Youngster[18], which had only four distinct controls). We performed leave-one-dataset-out and leave-one-disease-out cross-validation and calculated an AUC for each of the cross-validation testing results.

**Microbiome community analyses**. Alpha diversities were calculated based on the non-collapsed 100% OTU-level relative abundances, and included OTUs un-annotated at the genus level. We calculated alpha diversity metrics with the skbio. math.diversity.alpha.chao1, shannon, and simpson implementations.

We calculated the average abundance and ubiquity (Fig. 3c) of each genus as the mean of its average values in each data set across all patients with 16S data, regardless of their disease state. To calculate the abundance of each genus, we first calculated each genus's mean abundance within each data set. We counted only patients with non-zero abundance of the genus in this calculation. We then took the average of these mean abundances across all data sets. To calculate the ubiquity of each genus, we calculated the percent of patients with non-zero abundance of that genus in each data set. We then took the average of these ubiquities across all data sets.

**Code availability**. The code to reproduce all of the analyses in this paper is available at https://github.com/cduvallet/microbiomeHD. We encourage researchers to incorporate their existing and future case–control studies into the MicrobiomeHD database by contacting us.

**Data availability**. Raw sequencing data for each study can be accessed as described in Supplementary Table 3. The raw processed OTU tables can be accessed at the MicrobiomeHD database, available at https://doi.org/10.5281/zenodo.840333[16]. Supplementary Files, including the $q$ values for all genus-level comparisons in every data set, disease-associated genera for the diseases with more than three data sets, and a list of non-specific genera are available at https://github.com/cduvallet/ microbiomeHD. All other relevant data supporting the findings of the study are available in this article and its Supplementary Information files, or from the corresponding author on request.

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

## Acknowledgements

We thank the authors who made their data publicly available as well as those who provided data after personal communication. This project was supported in part by the MIT Center for Microbiome Informatics and Therapeutics, and C.D. acknowledges support by the Department of Defense through the National Defense Science & Engineering Graduate Fellowship (NDSEG) Program. We thank Shijie Zhao and Xiaoqian Yu for their early contributions to the literature review, as well as Nathaniel Chu and other members of the Alm and Irizarry groups for helpful discussions.

## Author contributions

C.D., T.G., and E.A. conceived of the research. C.D., S.G., and T.G. identified and downloaded the data sets. C.D. and T.G. wrote the code to process data. C.D. processed the data and performed all analyses. C.D., S.G., R.I., and E.A. interpreted the results and prepared the manuscript.

## Additional information

**Competing interests:** E.A. is on the Board of Directors of OpenBiome, a non-profit stool bank. E.A. is a co-founder of Finch Therapeutics, which aims to develop microbiome-based therapeutics. The remaining authors declare no competing financial interests.

