## [Peer Review File · Nature Communications]

Reviewers' comments:

Reviewer #1 (Remarks to the Author):

My research group reviewed the preprint version of this manuscript on May 18, 2017 and we prepared this joint review. The manuscript from Duvallet and colleagues seeks to create a database of case-control gut microbiome studies from 16S rRNA gene sequences and then use that database to look for consistent signatures of health and disease across a number of diseases. Overall, this is an interesting idea that is similar to an approach our lab and others have pursued to look across studies to identify signatures that are emblematic of lean or obese individuals. Unfortunately, the work has a number of technical problems and attempts to say too much without obtaining a full representation of data from the literature or incorporating the clinical nuances of the diseases they study.

Most of our concerns would overcome and the findings and impact of the paper would be greatly strengthened by testing the hypothesis that the core microbiome identified in Figure 3 is indeed sufficient enough to classify cases and controls across all of the studies. The authors should test the sensitivity/specificity of the core microbiome to successfully classify general disease and control cases across studies. If the hypothesis is that there is a core microbiome that is common to all diseases then the predictive accuracy of models using these core members should be relatively successful in finding generalized disease microbiomes regardless of whether it is cancer, obesity, diarrhea, etc. Even if the model is not generalizable across all diseases, it would be important to know whether a model for a single disease group is predictive of controls and cases within that disease group. Again, this was an approach that we used with Random Forest modeling to use one study to predict obesity status in other studies.

In the re-analysis of the CDI data, we are concerned that the non-CDI diarrheal controls have been grouped with the healthy controls (Figure 2). That seems to be the case at least for the study from our lab referenced- Schubert - as our study had 94 CDI cases, 89 diarrheal controls and 155 non-diarrheal controls. Thus the number of 243 controls (Table 1) would appear to represent a pooling of the diarrheal and non-diarrheal controls. We would suggest instead only using the diarrheal controls. We would strongly encourage the authors to confirm for each disease group that the control samples are similar across all studies.

Similar to the concern over the data used for the Schubert CDI data, the definition of 'cases' may need reconsidering for some diseases - what is a case for an HIV patient, for instance? Actively replicating virus? Reduced CD4 count? People that are HIV positive are often quite healthy with no detectable viral load. Similarly, IBD encompasses a range of bowel diseases and a 'case' of UC is different from a 'case' of Crohn's. Further, a note or clarification of whether any of these patients were on antibiotics (and if this could be a confounding factor) is necessary.

When testing and generating the 'core' microbiome across all diseases, we wonder whether testing for the core falsely amplified the CDI-related microbes in the pan-disease core because there were so many that were altered in cases? Similarly, we wondered how the authors could control for the variation in effect sizes when generating the list of core microbes? We would encourage something like a Z-transform like was used in the Sze and Schloss obesity meta-analysis

The ROC curves appeared to be inverted in the case of Dinh 2015 (supplemental figure 5). This can result from inverted categories, and can make the resulting AUC artificially appear lower than 0.5. We suggest the authors recalculate the AUCs from inverse ROC curves, so that all AUCs are between 0.5 and 1. In addition, the 0.5 AUC line only matters if one has an equal number of cases and controls - kappa - corrects for distribution in data. If 90% of the samples are cases, then one would expect to be correct 90% of the time, not 50.

Although the authors picked datasets that specifically dealt with the disease they were interested

in, there are a number of other datasets that were not included but could have easily been added. For example, we found 10 studies that included obesity data with their sequence data. The control samples used in the Schubert and Baxter studies from the Schloss lab could also be used to look at the effects of obesity.

The authors should also note that the samples used in the Zackular study are a subset of Baxter and so the studies are not independent.

Comments on overall writing style:

Overall we felt that the paper's organization and purpose is unclear. For instance, in the abstract, "Here, we introduce the MicrobiomeHD database, which includes 29 published case-control gut microbiome studies spanning ten different diseases" - Is this paper about the database? This database is hardly mentioned in the rest of the paper and the reader needs more details about how database was formed. Furthermore, the database does not appear to be comprehensive and there is no indication of whether the database will continue to be maintained over time. Alternatively, is the paper about the 'core' microbiome being able to predict healthy/disease? If so then it needs a direct test of this hypothesis. Is the paper about re-analyzing and confirming the findings of previous studies? Or by doing a true meta-analysis where the effect sizes are compared across studies? If either of those are true then the paper needs to be structured and concluded in a way that emphasizes those conclusions. The current version is a bit muddled in its structure and purpose.

It is oversimplifying the complex nature of the diseases to classify treatment therapies into antibiotics, probiotics or FMT. Each of these therapies can be affected by differences between human patients. At the very least, the discussion should include more caveats about these treatments particularly there are a few studies showing negative effects of antibiotic use and long-term development of colorectal cancer (Cao et al 2017) as well as studies showing manipulating the gut microbiome in mice via fecal transplants can spur the development of CRC tumors in a mouse tumorigenesis model (Zackular et al 2015).

Reviewer #2 (Remarks to the Author):

In this paper, Duvallet et al perform a meta-analysis of 29 published case-control human gut microbiome studies (16S rRNA targeted) in many disease contexts, and identify the degree to which microbiome differences are consistent within studies of the same disease or across diseases. Overall, I think that meta-analyses of this type are very valuable in that they help us to understand consistency across cohorts and parallels between different disease states. As such, this study provides some interesting insights such as that certain diseases like colorectal cancer are characterized by increase in pathogenic genera whereas others are characterized primarily by a depletion in host associated taxa. Also that certain diseases like obesity and HIV appear to have particular amount of disagreement between studies and really need more investigation. However, I think that there are some ways in which the analysis and presentation of the results could be improved that are detailed below.

Major comments.

- 1) The paper uses univariate analysis of genus level taxa as the primary read-out. As they state in the methods, using genera rather than OTUs allows for easier comparison between studies, but may not be optimal. Classifier-defined genera can represent varied degrees of complexity in terms

of their phylogenetic breadth, the number of gut taxa whose counts are “summed” within them, and the diversity of functional attributes of these individual taxa. I think that this limitation should be better discussed in the main paper and that some specific alterations detailed below should be made to better address this limitation. I am actually not convinced that this type of analysis could not be done at finer scales – such as by looking at mapping to greengenes OTUs.

- a. All OTUs not assigned at the genus level were discarded. How much did you discard?
 - b. Line 140 and line 185: a point is made about depletion of “butyrate-producing” Clostridiales – citing Lachnospiraceae and Ruminococcaceae. Some species within these very broad taxa produce butyrate but most do not. I would avoid statements that suggest that everything in a given genus have that functional attribute.
 - c. Line 187 “we also see an increase in prevalence of organisms” should say “we also see an increase in the prevalence of genera that contain organisms”
 - d. Lines 220-222. The authors describe genera that were both health and disease associated saying it could be due to confounders, technical artifacts, or these organisms playing different roles in different disease or community contexts. Another explanation may be that there are both health associated and disease associated species/strains that are in the same genus, and that your genus in the different studies is not actually representing the same organism.
- 2) The paper perhaps over simplifies the complexities of the different diseases surveyed and how the individual studies of the “same disease” may have differences in what the disease state actually represents. For instance, for HIV, it may make a big difference if individuals are on ART or not and whether they have low or relatively high CD4 positive T cell counts. The cohort in Dinh et al is entirely on ART and in the Parades and Lozupone et al papers have a mix of untreated and treated. For autism, certain studies focus on kids with GI involvement and others do not. There are different forms of IBD that have been shown to have varying degrees of phenotype in feces – e.g. Ileal Crohn’s disease is more than colonic in some cases. I think that this paper needs to better acknowledge that many of these individual papers did not simply treat these cohorts as “yes” “no” with disease.
- 3) Related to point 2 above, On lines 310-314, the authors discuss that there are “false positives” in other studies that were not found in this re-analysis, citing that they could not detect an association between Prevotella and autism or rheumatoid arthritis in their reanalysis. I think that I would need more supporting info on how this analysis differed from those conducted in these original papers before I would believe that the reports in the original papers were reporting false positives. The original papers may have better dealt with the complexities of their individual disease systems (e.g. controlled for potential confounders), which was not done here.
- 4) I liked the concept of establishing “core” health and disease associated microbes but I did not think that the definition was rigorous enough. Can a genus that “is significantly enriched or depleted in at least one dataset from at least 2 different diseases)” really be considered a “core”?
- 5) Line 311. It says here that Prevotella was reported to be enriched with autism but this is not true. It was reported to be depleted with autism.
- 6) Lines 325-327: the point is made that “Individual studies are plagued by low statistical power, confounding variables etc.” Isn’t that also a weakness of the analysis conducting here since you are just re-analyzing all of the original studies and seeing where they agree? One way to actually increase power would be to apply Fisher’s method (https://en.wikipedia.org/wiki/Fisher%27s_method).
- 7) Figures 8 and 9 are pretty useless without any information attached on what genera the rows are representing

Minor comments

1) typos/grammer

a. Page 4 line 68: Transplantation is misspelled

2) Figure 4 would be easier to look at if healthy was always on the left and disease on the right (or vice versa). Why was only Shannon index looked at?

- 3) Lines 270-272. "abundance" should be replaced with "relative abundance" throughout since 16S does not provide quantitative information.
- 4) In figures 2 and 3 – it is not shown what genera are being represented. I see now that this information is in Figure S6 and S7, but I did not realize this when reading the paper because they are not referenced in the main text (just at the end of a figure legend).

Summary of the response

We thank the reviewers for their critical reading of our manuscript and thoughtful comments. As requested by the editor, our response includes major revisions in response to comments from the reviewers.

Specifically, we:

- tested the hypothesis that there is a core microbiome which can differentiate health from general disease, and discuss these results in the main text and supplement;
- analyzed subsets of case patients separately for diseases in which heterogenous patient groups had previously been combined (e.g. IBD, CDI, LIV, and ART). These results are discussed in the main text and included in the supplement; and
- added more nuanced discussion of the complexities of the different diseases surveyed and of the interpretation of the possible treatment strategies suggested by the different types of microbiome shifts.

We respond to each reviewer point-by-point below, and have also included in our submission a manuscript file with all revisions tracked.

Reviewer #1

Reviewer #1's main concerns related to whether the core microbial response to disease generalized across diseases. They indicated that most of their concerns could be alleviated by more directly testing the hypothesis that there is a core microbial response that is common to all diseases, and suggested building a classifier to classify general cases from controls across all datasets. Reviewer #1 was also concerned that some of our coarse-grained analyses and conclusions did not adequately consider the heterogeneities and complexities of the diseases surveyed. Finally, Reviewer #1 raised concerns about our combination of heterogenous patient groups, especially in the IBD datasets and their CDI dataset (Schubert et al.).

We performed the healthy vs. disease classifier analysis requested and present these new results in the revised manuscript. We have also edited our language throughout the text to

better acknowledge the complexities and heterogeneities of the surveyed diseases. Additionally, we modified our analysis of the CDI Schubert dataset to include both healthy vs. CDI and healthy vs. non-CDI diarrhea comparisons and include this new comparison in the main text. Finally, we perform a new, stratified analysis for the IBD datasets and present these results in the supplement.

Reviewer #1 (Remarks to the Author):

My research group reviewed the preprint version of this manuscript on May 18, 2017 and we prepared this joint review. The manuscript from Duvall et al. seeks to create a database of case-control gut microbiome studies from 16S rRNA gene sequences and then use that database to look for consistent signatures of health and disease across a number of diseases. Overall, this is an interesting idea that is similar to an approach our lab and others have pursued to look across studies to identify signatures that are emblematic of lean or obese individuals. Unfortunately, the work has a number of technical problems and attempts to say too much without obtaining a full representation of data from the literature or incorporating the clinical nuances of the diseases they study.

Most of our concerns would be overcome and the findings and impact of the paper would be greatly strengthened by testing the hypothesis that the core microbiome identified in Figure 3 is indeed sufficient enough to classify cases and controls across all of the studies. The authors should test the sensitivity/specificity of the core microbiome to successfully classify general disease and control cases across studies. If the hypothesis is that there is a core microbiome that is common to all diseases then the predictive accuracy of models using these core members should be relatively successful in finding generalized disease microbiomes regardless of whether it is cancer, obesity, diarrhea, etc. Even if the model is not generalizable across all diseases, it would be important to know whether a model for a single disease group is predictive of controls and cases within that disease group. Again, this was an approach that we used with Random Forest modeling to use one study to predict obesity status in other studies.

- **Author response:** The reviewer brings up an intriguing point about additional ways to test the generalizability of the core microbiome. We re-built each study's random forest classifiers but limited the features to just the 64 genera which we identified as core. We found that these classifiers performed as well as the original classifiers, which had included all genera as features (Response Figure 1).

Response Figure 1. Classifiers built from features limited to “core” genera (y-axis) perform as well as classifiers built on all genera in each dataset (x-axis) (Pearson $r = 0.98$). AUC, area under the ROC curve.

- To further address the question of whether we could find a robust, generalized signal for diseased microbiomes regardless of the disease type, we built two additional classifiers to distinguish healthy patients from any type of case patient. We had not originally performed this analysis due to concerns that batch effects between datasets would obscure any biological signals. However, at the prompting of Reviewer #1, we built a healthy vs. disease classifier. Surprisingly, we found that in spite of the technical and biological heterogeneities across studies, a generalized microbial response could still be identified across datasets and diseases (Supplementary Figure 13). This is an intriguing putative confirmation of our hypothesis that there is a generalized microbial response to disease.
- In this paper, our intention was to identify microbes that respond non-specifically to disease as a caution to researchers that the majority of associations identified in any single study may not be specific to the disease they are studying. Therefore, we chose to report these intriguing but still preliminary results in the supplementary material, rather than emphasize them in the main text (lines 247-253 and Section 6.4: “Healthy vs. disease classifier identifies general microbiome shifts”).
- Our heuristic definition identifies microbes which respond non-specifically to disease by finding those which are significantly associated with more than one disease. Future case-control studies can use the knowledge that many microbes respond non-specifically to disease to narrow down putative causal or diagnostic biomarkers which fall outside of the core response and are thus more likely to be specific to the disease being studied. The healthy vs. disease classifier adds strength to the underlying

biological hypothesis of a core response, but is a less actionable result than the list of 'core' microbial responses.

Supplementary Figure 13: *Left:* leave-one-dataset-out classifier. *Both x-axes:* the AUC from each dataset's single classifier. *y-axis:* the AUC of a classifier trained on all other datasets to distinguish healthy from unhealthy patients, tested on the left out dataset. *Right:* leave-one-disease-out classifier. *y-axis:* AUC from a classifier trained to distinguish healthy from unhealthy patients on all datasets except those of the tested disease. AUCs for each dataset were built from the classification probabilities on each test sample.

In the re-analysis of the CDI data, we are concerned that the non-CDI diarrheal controls have been grouped with the healthy controls (Figure 2). That seems to be the case at least for the study from our lab referenced-Schubert - as our study had 94 CDI cases, 89 diarrheal controls and 155 non-diarrheal controls. Thus the number of 243 controls (Table 1) would appear to represent a pooling of the diarrheal and non-diarrheal controls. We would suggest instead only using the diarrheal controls. We would strongly encourage the authors to confirm for each disease group that the control samples are similar across all studies.

- **Author's response:** We had originally combined the diarrheal and non-diarrheal controls in the Schubert study in order to isolate the effect of CDI diarrhea, but agree that using only non-diarrheal patients as controls is a stronger analysis. We split our analysis of the CDI Schubert dataset into two separate diarrheal comparisons: CDI diarrhea vs. healthy and non-CDI diarrhea vs. healthy. We were happy to see that diarrheal patients responded similarly regardless of diarrhea etiology, and have included this analysis in the main text and figures (manuscript Figures 1 and 2).
- For all other studies, we used the originally-defined control patients as controls in our re-analysis. In all but two studies, which used non-IBD patients as controls (Gevers et al. (2014) and Papa et al. (2012)), these were healthy patients. Because we performed all of our analyses on each dataset independently rather than combining the raw data, we believe that including these analyses in our comparisons is valid even though the control patients are not strictly healthy. While not ideal, incorporating more studies increases our power to identify consistent disease-associated signals, and the increased power from including these two studies far outweighs the slight discrepancy in control groups.

Similar to the concern over the data used for the Schubert CDI data, the definition of 'cases' may need reconsidering for some diseases - what is a case for an HIV patient, for instance? Actively replicating virus? Reduced CD4 count? People that are HIV positive are often quite healthy with no detectable viral load. Similarly, IBD encompasses a range of bowel diseases and a 'case' of UC is different from a 'case' of Crohn's.

- **Author's response:** We agree with the reviewer that disease heterogeneities and complexities are important considerations. Our primary goal with this study was to synthesize existing knowledge and determine whether we could draw generalizable insights about microbial associations with disease, in spite of the heterogeneity between conditions, patient cohorts, study designs, and experimental methods. In all cases, we used the original authors' definitions of case patients for our re-analysis. We did collapse certain case groups together (e.g. UC and CD) so that we could improve our power and learn generalized patterns of health and disease. We recognize that not all 'case' patients are the same, but believe that there is still merit to comparing them in order to identify and extract consistent patterns of dysbiosis, if there are any. In fact, individual case-control studies essentially do this as well, coming to general conclusions about diseases from their unique, non-representative patient cohorts. However, we acknowledge that our original language and analysis was perhaps too broad and did not properly address these important subtleties, and have edited the main text accordingly (e.g. lines 155-157 and 187-197).
- Furthermore, we now provide a stratified analysis of IBD patients (controls vs. ulcerative colitis and controls vs. Crohn's disease) in the supplement. Both IBD subtypes showed similar qualitative patterns of disease-associated shifts (e.g. depletion of health-associated bacteria and moderate classifiability). Our main argument in the paper is that one can identify qualitative shifts which are consistent across datasets of the same disease. Because this shift is the same for both IBD subtypes, we continue to group CD and UC patients together as IBD cases in the main text and have added this expanded analysis to the supplement (lines 155-158, Supplementary Figures 7 and 8, and Section 6.2: "Stratifying heterogenous case groups shows consistent disease-specific signals").
- Finally, by making the MicrobiomeHD database of case-control studies available online, we enable future researchers to do subsequent analyses, stratifying any of these disease states even further.

Further, a note or clarification of whether any of these patients were on antibiotics (and if this could be a confounding factor) is necessary.

- **Author Response:** We agree with the reviewer that it is important to describe potential confounders for each study. For example, some of the IBD patients were indeed on antibiotics, which is an obvious factor that might shape their gut microbiome. However, for the studies included in this analysis, we either lacked sufficient metadata or we lacked a sufficient number of samples within each disease/treatment category to

quantitatively assess how antibiotics affect the results. For example, only 6 of the 28 studies included had useful antibiotics metadata, and of these, only 1 (CDI Schubert) had more than 5 controls who were on antibiotics. Thus, performing a truly stratified analysis was not possible, and we must make the best use of the data collected in the original studies. Future case control studies should focus on better separating treatment and disease variables by collecting detailed metadata on antibiotic and other medication usage, and perhaps also by recruiting controls undergoing a variety of treatments. We have added discussion of this confounder in the main text (lines 187-197).

When testing and generating the 'core' microbiome across all diseases, we wonder whether testing for the core falsely amplified the CDI-related microbes in the pan-disease core because there were so many that were altered in cases?

- **Author response:** The reviewer brings up an excellent point about the robustness of our 'core' heuristic considering that many microbes are altered in diarrhea studies. Indeed, it is possible that rather than defining a core response to disease, our 'core' microbes are rather representing a core response to diarrhea.
- To test this hypothesis, we re-defined the 'core' genera based on all datasets except diarrheal ones (CDI Schubert, EDD Singh, CDI Vincent, and CDI Youngster). Reassuringly, we found that 40 out of the 67 original 'core' genera were recovered (Supplementary Figure 12). Thus, the majority of the core microbiome is robust to the exclusion of diarrhea datasets. This analysis has been included in the supplement (lines 236-238, Supplementary Figure 12, and Supplementary Section 6.3: "Core microbial response is robust to different definitions").

Supplementary Figure 12. The majority of core microbes are robust to the exclusion of diarrhea datasets from consideration. The bottom-most bar shows order-level phylogeny, colored as in Figure 3A of the main paper. The top bar of the heatmap shows the original core microbes, including all datasets. The middle bar shows the re-defined core microbes after excluding all diarrhea datasets. The bottom bar of the heatmap shows the core microbes defined using Stouffer's method (Stouffer combined $q < 0.05$). [Stouffer method results added by request of Reviewer #2]

Similarly, we wondered how the authors could control for the variation in effect sizes when generating the list of core microbes? We would encourage something like a Z-transform like was used in the Sze and Schloss obesity meta-analysis

- We explicitly show the effect sizes for each genus in each dataset in Supplementary Figure 15, and now include these values in Supplementary File S5 (<https://github.com/cduvallet/microbiomeHD/tree/master/final/supp-files>). We did not find explicit mention of z-scores or z-transformations in the Sze and Schloss paper, but would be happy to include an alternative effect size analysis in the supplement if the reviewers recommend a different method.
- We used non-parametric approaches throughout all of our analyses to avoid any need to make assumptions about the distribution of the data, as any Z-transform would require. Such assumptions would almost certainly be violated due to the batch effects between studies, and it is not in the scope of this paper to develop robust methods to approach these effects. Future analyses will certainly need to use more rigorous methods to account for effect size and batch effects, especially if they address microbiome shifts across studies at finer resolutions (e.g. at the OTU level).

The ROC curves appeared to be inverted in the case of Dinh 2015 (supplemental figure 5). This can result from inverted categories, and can make the resulting AUC artificially appear lower than 0.5. We suggest the authors recalculate the AUCs from inverse ROC curves, so that all AUCs are between 0.5 and 1.

- **Author's response:** We also found the low AUC result for the Dinh et al. study surprising. We double-checked the disease labels to ensure that the low AUC was not an artifact of mislabeling. We confirmed that our case labels matched those provided in the raw data (in the SRA Run Table column 'disease': <https://www.ncbi.nlm.nih.gov/Traces/study/?acc=SRP039076>). The low AUC may be an artifact of low sample size: this study has one of the smallest sample sizes in our collection (36 samples), and would have had only 3 healthy and 4 case samples in each cross-validation fold. We also find that the low AUC result is consistently low even as we vary the classifier parameters (Supplementary Figures 16 and 17), which indicates that there may be some underlying biology which is confusing the classifier. For example, it is possible that patients who have very similar microbiomes but who are discordant for disease are spread between training and test sets, leading to consistently worse-than-chance classification on the test set. Considering the small sample size, the lack of univariate associations with HIV, and the demonstrated environmental/behavioral confounders in other HIV datasets, we believe that this classifier is probably picking up some non-HIV signal that is making it perform worse than expected, and that calculating the AUC from the inverted ROC curve is not a legitimate fix to the underlying issue. As we mention in the main text, HIV is one example of a complex disease with many possible confounders, which will require more studies and especially larger studies to tease apart.

In addition, the 0.5 AUC line only matters if one has an equal number of cases and controls - kappa - corrects for distribution in data. If 90% of the samples are cases, then one would expect to be correct 90% of the time, not 50.

- **Author response:** The ROC curves show the true positive and false positive rates of the classifier at varying classification probability thresholds, not the percent accuracy. The $y = x$ line (corresponding to an AUC of 0.5) illustrates the behavior of a random classifier at different probability cutoff thresholds (e.g. one that randomly predicts 50% of samples as cases as well as one that randomly predicts 90% of samples as cases, etc). In an example with 90 cases and 10 controls and a classifier that randomly predicts a sample as a case 50% of the time, it would predict 45 out of 90 true cases as cases and 5 out of 10 true controls as cases. This is equivalent to a false positive rate of 0.5 and a true positive rate of 0.5 as well, which is the (0.5, 0.5) point on the ROC curve. Thus, the ROC curve (and resulting AUC calculated from it) are insensitive to class imbalances. [<https://doi.org/10.1016/j.patrec.2005.10.010>]

Although the authors picked datasets that specifically dealt with the disease they were interested in, there are a number of other datasets that were not included but could have easily been added. For example, we found 10 studies that included obesity data with their sequence data. The control samples used in the Schubert and Baxter studies from the Schloss lab could also be used to look at the effects of obesity.

- **Author's response:** Our primary goal with this study was to re-analyze datasets in order to extract generalizable patterns from previously studied microbial disease associations. For this to be a feasible endeavor and in order to remain consistent across studies and patient cohorts, we decided to limit our analyses to the original scopes of the papers. We agree that many additional comparisons could have been performed and more studies included, but decided that reporting this preliminary meta-analysis would be a more valuable contribution to the field than a fully expansive analysis. Additionally, many studies with reported obesity or other disease metadata proved to have difficult-to-access raw data or metadata, and in other cases did not have balanced obese/non-obese samples to make good comparisons. Furthermore, analyzing obesity patients in non-obesity studies would have added an additional confounder from each patient's primary disease state. Finally, previous papers - including the Schloss meta-analysis - have shown little effect of obesity on the microbiome, and so we were less interested in pursuing every possible comparison for obesity specifically.
- As more studies are published and data for more diseases becomes readily available, we hope that the methods and conceptual frameworks presented in this paper can guide future, more comprehensive meta-analyses. We also hope that our study is just the beginning of a new standard in the field, where each individual case control study's findings are compared to the existing body of work. In these cases, we agree that it will be useful to consider secondary metadata from published case control studies (e.g. obesity data from a non-obesity-focused study). We have compiled all of the publicly available metadata from these studies and their associated OTU tables in our MicrobiomeHD database, so that future researchers and clinicians can more easily perform these meta-analyses.

The authors should also note that the samples used in the Zackular study are a subset of Baxter and so the studies are not independent.

- **Author response:** We thank the reviewer for pointing out that the Zackular and Baxter studies are not independent. We have removed the smaller of the two, Zackular 2016, from our analyses.

Comments on overall writing style:

Overall we felt that the paper's organization and purpose is unclear. For instance, in the abstract, "Here, we introduce the MicrobiomeHD database, which includes 29 published case-control gut microbiome studies spanning ten different diseases" - Is this paper about the database? This database is hardly mentioned in the rest of the paper and the reader needs more details about how database was formed. Furthermore, the database does not appear to be comprehensive and there is no indication of whether the database will continue to be maintained over time. Alternatively, is the paper about the 'core' microbiome being able to predict healthy/disease? If so then it needs a direct test of this hypothesis. Is the paper about re-analyzing and confirming the findings of previous studies? Or by doing a true meta-analysis where the effect sizes are compared across studies? If either of those are true then the paper needs to be structured and concluded in a way that emphasizes those conclusions. The current version is a bit muddled in its structure and purpose.

- **Author response:** The main finding of this paper is that there is a 'core' microbial response to health and disease, and that the majority of individual disease-associated bacteria also respond non-specifically to multiple diseases. Furthermore, we show that there are characterizable shifts in the microbiome that are consistently associated with certain diseases, in spite of heterogeneity between patient cohorts and datasets. Our meta-analysis does not focus on directly comparing effect sizes of individual bacteria across studies because technical issues (i.e. batch effects) need to be resolved before such comparisons can be convincingly tied to certain disease states. However, we believe that our framework for characterizing dysbioses and identifying non-specific microbial responses is a significant contribution to the field. Furthermore, our collection and standardized re-analysis fits into the traditional definition of a meta-analysis, which is to "find the knowledge in the [abundance of] information" [Glass, G. V. (1976). Primary, secondary, and meta-analysis of research. *Educational researcher*, 5(10), 3-8]. We have edited the abstract to more directly clarify the main points of our paper.
- We have presented the MicrobiomeHD database so that future researchers can perform their own similar analyses, which is an important take-away from our paper. Although not the scientific focus of our paper, we include the database in the abstract so that even readers who only skim the paper are aware of this consolidated resource. We hope that as the field of microbiome science develops and matures, more established and well-supported resources will take the role of the MicrobiomeHD database. However, data sharing and data standards are not currently the norm, and even our relatively limited collection of 28 datasets required significant time and energy to curate and re-process. Thus, having all of the OTU tables processed with the same methods and with the associated metadata available in the MicrobiomeHD database is an important contribution and will hopefully be useful to future researchers. While data storage and processing standards are being established in the field, we are also happy to process future case-control datasets and include them in MicrobiomeHD if the investigators wish to compare their results with the existing datasets. We have clarified this intention on the

MicrobiomeHD Zenodo page (zenodo.org/record/569601) and in the paper's Methods, adding the following sentence (lines 526-528): "We encourage researchers to incorporate their existing and future case-control studies into the MicrobiomeHD database by contacting us."

It is oversimplifying the complex nature of the diseases to classify treatment therapies into antibiotics, probiotics or FMT. Each of these therapies can be affected by differences between human patients. At the very least, the discussion should include more caveats about these treatments particularly there are a few studies showing negative effects of antibiotic use and long-term development of colorectal cancer (Cao et al 2017) as well as studies showing manipulating the gut microbiome in mice via fecal transplants can spur the development of CRC tumors in a mouse tumorigenesis model (Zackular et al 2015).

- **Author response:** We agree that classifying treatment strategies into antibiotics, probiotics, and FMT is an oversimplification. We have modified our discussion to clarify that our framework for describing dysbiosis is intended as a hypothesis-generating tool, rather than a definitive roadmap for treatment. We wish to highlight that qualitative groupings of microbiome responses across diseases will likely define how we approach the development of treatment strategies, rather than suggest actual treatment therapies for these diseases. We have added the following text to the discussion (lines 384-389): "Furthermore, characterizing "dysbioses" by their directionality and extent is a useful framework to generate hypotheses for future microbiome research on complex, heterogenous diseases. While many of these conditions are unlikely to be fully treated by antibiotics, probiotics, or FMTs, our proposed framework could guide the search for new therapies and etiologies by generating testable hypotheses with higher likelihoods of success."

Reviewer #2

Reviewer #2's main concern was related to our oversimplification of the complexities of different diseases. Specifically, Reviewer #2 encouraged us to better acknowledge that many of the original papers did not consider many of these diseases as blanket cases and controls. They also encouraged us to more explicitly discuss possible patient confounders such as medication usage. Reviewer #2 also commented on our 'core'-defining heuristic definition, and specifically questioned whether it was rigorous enough. Finally, reviewer #2 discussed the limitation that our analyses were carried out at the genus level.

As discussed above in the response to Reviewer #1, we have edited our language throughout the manuscript to better acknowledge the complexities of different diseases. Furthermore, we performed new, stratified analyses of patients subtypes in one CDI study (CDI Schubert, in the main text), all of the IBD studies (in the supplement), and the liver and arthritis studies (in the supplement). We also added discussion of possible confounders and other factors contributing to patient heterogeneity in the main text. In response to the comment on our 'core' heuristic, we now include a comparative analysis of our heuristically-defined 'core' microbes with those defined by a more traditional statistical method, and add additional discussion and justification

of our heuristic definition over other methods. Finally, we incorporated Reviewer #2's suggested clarifications to the text to prevent over-generalizing our results beyond the genus level. We also incorporated their many helpful and detailed comments on clarifications to make to the figures and text.

Reviewer #2 (Remarks to the Author):

In this paper, Duvallet et al perform a meta-analysis of 29 published case-control human gut microbiome studies (16S rRNA targeted) in many disease contexts, and identify the degree to which microbiome differences are consistent within studies of the same disease or across diseases. Overall, I think that meta-analyses of this type are very valuable in that they help us to understand consistency across cohorts and parallels between different disease states. As such, this study provides some interesting insights such as that certain diseases like colorectal cancer are characterized by increase in pathogenic genera whereas others are characterized primarily by a depletion in host associated taxa. Also that certain diseases like obesity and HIV appear to have particular amount of disagreement between studies and really need more investigation. However, I think that there are some ways in which the analysis and presentation of the results could be improved that are detailed below.

Major comments.

1) The paper uses univariate analysis of genus level taxa as the primary read-out. As they state in the methods, using genera rather than OTUs allows for easier comparison between studies, but may not be optimal. Classifier-defined genera can represent varied degrees of complexity in terms of their phylogenetic breadth, the number of gut taxa whose counts are "summed" within them, and the diversity of functional attributes of these individual taxa. I think that this limitation should be better discussed in the main paper and that some specific alterations detailed below should be made to better address this limitation. I am actually not convinced that this type of analysis could not be done at finer scales – such as by looking at mapping to Greengenes OTUs.

- **Author Response:** We agree with the reviewer that our genus-level summary of the data limits the phylogenetic resolution of our results. However, it was a tradeoff between phylogenetic resolution and systemic batch effects that plague 16S sequencing data. At the OTU level, batch effects are highly prominent, but these effects are reduced by binning at the genus level. Batch effects are prominent even when mapping only to Greengenes representative sequences (i.e. closed reference clustering). For example, different regions of the 16S gene (e.g. V2 vs. V4) are more or less efficient at capturing particular taxonomic groups, and these biases are strongest at the OTU level. As suggested by the reviewer, we have added the following text to our results section (lines 93-98): "By collapsing data to the genus level, we lost the sensitivity to detect fine-scale differences in species or strain abundances across case and control groups, but we minimized certain batch effects that plague comparisons across studies. Thus, we took a course-grained approach to optimize our ability to compare data across studies at the expense of phylogenetic resolution."

a. All OTUs not assigned at the genus level were discarded. How much did you discard?

- **Author’s response:** On average, we discarded a minimum of 3% of reads (T1D Alkanani) and a maximum of 36% of reads (PAR Scheperjans). All but two of our 28 datasets had over 80% of their reads annotated to the genus level (Response Figure 2).

Response Figure 2. Percent of data which was annotated to genus level. Bars show the mean total abundance across all samples, with bootstrapped 95% confidence intervals (as implemented in seaborn’s barplot plotting function).

b. Line 140 and line 185: a point is made about depletion of “butyrate-producing” Clostridiales – citing Lachnospiraceae and Ruminococcaceae. Some species within these very broad taxa produce butyrate but most do not. I would avoid statements that suggest that everything in a given genus have that functional attribute.

- **Author Response:** The reviewer is correct that not all members of these taxonomic groups are known to produce butyrate. However, the dominant *Lachnospiraceae* (e.g. *Eubacterium* and *Rosburia*) and *Ruminococcaceae* (e.g. *Faecalibacterium*) genera in the gut harbor butyrate-production genes (<http://onlinelibrary.wiley.com/doi/10.1111/j.1462-2920.2009.02066.x/full>). In order to clarify this point, we have altered line 151 to read: “While not all genera within *Ruminococcaceae* and *Lachnospiraceae* are verified SCFA producers, the dominant genera within these families are known to harbor genes for short chain fatty acid production and are often associated with colonic health.”

c. Line 187 “we also see an increase in prevalence of organisms” should say “we also see an increase in the prevalence of genera that contain organisms”

- **Author Response:** We have made the suggested change to the text.

d. Lines 220-222. The authors describe genera that were both health and disease associated saying it could be due to confounders, technical artifacts, or these organisms playing different roles in different disease or community contexts. Another explanation may be that there are both health associated and disease associated species/strains that are in the same genus, and that your genus in the different studies is not actually representing the same organism.

- **Author Response:** The reviewer brings up an excellent point. We have added the following sentence to that section (lines 245-247): “Alternatively, different species/strains within these genera may play alternate roles across diseases or community contexts, giving rise to variable responses at the genus level.”

2) The paper perhaps over simplifies the complexities of the different diseases surveyed and how the individual studies of the “same disease” may have differences in what the disease state actually represents. For instance, for HIV, it may make a big difference if individuals are on ART or not and whether they have low or relatively high CD4 positive T cell counts. The cohort in Dinh et al is entirely on ART and in the Parades and Lozupone et al papers have a mix of untreated and treated. For autism, certain studies focus on kids with GI involvement and others do not. There are different forms of IBD that have been shown to have varying degrees of phenotype in feces – e.g. Ileal Crohn’s disease is more than colonic in some cases. I think that this paper needs to better acknowledge that many of these individual papers did not simply treat these cohorts as “yes” “no” with disease.

- **Author Response:** The reviewer brings up a good point about the subtleties of disease subtypes and cohorts. Some amount of coarse-graining was necessary for our meta-analyses. We combined certain conditions based on disease similarity for our cross-disease comparison (e.g. CD and UC). We did provide some discussion of these diseases in the original supplemental text (Supplementary Section 6.1). We also discussed differences between HIV cohorts across the three studies, which drove spurious associations within certain studies (see paragraph in Section 2.2 starting with ‘In some studies, confounding variables may drive associations’). As suggested by the reviewer, we have now included discussion of ART treatment status as a potential confounder in the HIV summary paragraph in the supplemental text (Supplementary Section 6.1.5, lines 856-866).
- Furthermore, we now provide a stratified analysis of IBD patients (controls vs. ulcerative colitis and controls vs. Crohn’s disease). Both IBD subtypes were similarly classifiable, and the disease-associated shifts in both CD and UC are similarly marked by a depletion of health-associated bacteria. Because our main finding is that qualitative patterns of disease-associated shifts are consistent across studies of the same disease, and both UC and CD show the same type of shift (i.e. depletion of health-associated bacteria), we have continued to group CD and UC patients together as IBD cases in the main text and include this expanded analysis in the supplement (Supplementary Figures 7 and 8 and Section 6.2). We also provide stratified analyses of the liver (MHE and liver cirrhosis) and arthritis (rheumatoid and psoriatic arthritis) datasets, and similarly find little difference between the qualitative shifts observed in the grouped and stratified patient comparisons.

- Our goal with this study was to identify whether broad patterns of dysbiosis can be found across a variety of heterogeneous patient cohorts, study designs, and experimental methods. As stated above in response to Reviewer #1, we believe that comparing across heterogeneous 'case' groups is a valid and useful way to identify and extract consistent patterns of dysbiosis, if there are any. Existing case-control studies essentially treat their 'case' cohorts as representative of the broader disease state, drawing general conclusions from their individual, non-representative patients. As discussed above, we have added additional discussion of the patient heterogeneities in the main text (lines 155-157, 187-197, and Supplementary Section 6.2), and we have made the data available in the MicrobiomeHD database so that others can perform any more stratified analyses that they wish.

3) Related to point 2 above, On lines 310-314, the authors discuss that there are "false positives" in other studies that were not found in this re-analysis, citing that they could not detect an association between *Prevotella* and autism of rheumatoid arthritis in their reanalysis. I think that I would need more supporting info on how this analysis differed from those conducted in these original papers before I would believe that the reports in the original papers were reporting false positives. The original papers may have better dealt with the complexities of their individual disease systems (e.g. controlled for potential confounders), which was not done here.

- **Author Response:** We agree with the reviewer that our suggestion of false positives may be overly strong. Our methods were often more conservative than those employed in the original studies (e.g. non-parametric statistics; OTU data summarized at the genus level), which may have resulted in the loss of subtle, but significant associations between the microbiome and disease. We discussed this in the supplemental information, but have now changed the main text to read 'potential false positives.' We have also added the following sentence to this section of the manuscript (lines 344-347): "For each of these diseases, the associations with *Prevotella* were weakly significant or complicated by confounding factors. In our more statistically conservative re-analysis, we found no association between autism or arthritis and *Prevotella*."

4) I liked the concept of establishing "core" health and disease associated microbes but I did not think that the definition was rigorous enough. Can a genus that "is significantly enriched or depleted in at least one dataset from at least 2 different diseases)" really be considered a "core"?

- **Author response:** Our motivation for identifying a 'core' microbial response to disease was to highlight the fact that many genera which are significantly associated with a given disease are actually non-specifically responding to multiple diseases. Finding one bacteria which is associated with multiple diseases casts doubts on that bacteria being a direct response or cause of that disease, and rather suggests that it is simply a part of a general dysbiosis. Currently, published case-control studies work off of the assumption that the bacterial associations they find are indicative of something specific to their disease of interest. In contrast, the 'core' microbial response that we present highlights that many genera have non-specific responses, and that these non-specific responders should be excluded when identifying bacteria for follow up experiments or diagnostics.

- We defined core microbes as those which were significantly enriched or depleted in two diseases because it was the simplest, most direct way to identify non-specific responders. Our analyses throughout are very conservative, and we believe that finding the same significant microbe (at an FDR $q < 0.05$) in two different diseases is worth noting, especially considering how few significant genera each individual dataset tends to have. We chose not to use Fisher's or Stouffer's methods to combine q-values across datasets to identify our 'core' responders because these methods are less appropriate to answer our biological question of interest. With these methods, one highly significant genus in one large dataset could dominate over other datasets' mediocre p-values and be flagged as 'core', despite only being associated with one disease.
- Nevertheless, we have added a comparison of the results from our 'core' heuristic with the results from combining q-values using Stouffer's method in the supplement (Supplementary Figure 12 and Supplementary Section 6.3: "Core microbial response is robust to different definitions"). When we use Stouffer's method to define core genera, we find that the two methods do not provide conflicting results (i.e. one does not define a "beneficial" core genus while the other defines it as "harmful") but that the Stouffer method finds many more core genera than our heuristic. Thus, our heuristic definition is more directly related to our biological question and also yields more conservative results.

5) Line 311. It says here that Prevotella was reported to be enriched with autism but this is not true. It was reported to be depleted with autism.

- **Author response:** We thank the reviewer for pointing out this mistake, and have corrected the text accordingly (line 344).

6) Lines 325-327: the point is made that "Individual studies are plagued by low statistical power, confounding variables etc." Isn't that also a weakness of the analysis conducting here since you are just re-analyzing all of the original studies and seeing where they agree? One way to actually increase power would be to apply Fisher's method (https://en.wikipedia.org/wiki/Fisher%27s_method).

- **Author's response:** While each individual study is indeed limited by its original weaknesses, comparing the results from each independent study increases our power to detect putatively true effects, regardless of the method used for the comparison. In fact, the conceptual basis of Fisher's method (and the similar Stouffer's method) is that features which are significant in more than one independent study carry more statistical weight. Our core microbiome heuristic and other comparisons throughout the paper are essentially taking advantage of this same concept: signals which are present in multiple independent studies of the same disease are more likely to be real disease-associated microbiome perturbations than ones which are only found in one study. As discussed above in response to point #4, we find that our core-defining heuristic is a more direct way to identify microbes which respond non-specifically to multiple diseases, but we have also included the Stouffer's method analysis results in the supplement (Supplementary Section 6.3).

7) Figures 8 and 9 are pretty useless without any information attached on what genera the rows are representing

- **Author response:** We have included the genus information in the updated figures. We thank the reviewer for this useful recommendation (Supplementary Figures 14 and 15).

Minor comments

1) typos/grammer

a. Page 4 line 68: Transplantation is misspelled

- **Author response:** We thank the reviewer for pointing out this mistake, and have corrected the text accordingly.

2) Figure 4 would be easier to look it if healthy was always on the left and disease on the right (or vice versa). Why was only Shannon index looked at?

- **Author response:** We have made the suggested change, and included two additional metrics of alpha diversity. The Shannon Index is essentially a weighted combination of richness and evenness, and so we considered it as a good summary for alpha diversity. The reviewer brings up a good point that there are many other ways to define alpha diversity. We have included figures showing evenness (Simpson's) and richness (Chao1) in the supplement.

3) Lines 270-272. "abundance" should be replaced with "relative abundance" throughout since 16S does not provide quantitative information.

- **Author response:** We have made the suggested changes.

4) In figures 2 and 3 – it is not shown what genera are being represented. I see now that this information is in Figure S6 and S7, but I did not realize this when reading the paper because they are not referenced in the main text (just at the end of a figure legend).

- **Author response:** We have added references to the labeled figures into the main text of the paper, when Figures 2 and 3 are first introduced for each disease type.

Reviewers' comments:

Reviewer #2 (Remarks to the Author):

I feel that the paper has been improved based on the responses to the Reviewer's comments. However, there were a couple of things in their rebuttal that I did not quite agree with:

1) Reviewer 1 made the point that in the re-analysis of the CDI data, that the non-CDI diarrheal controls were grouped with the healthy controls, and that they should confirm that for each disease group that the control samples are similar across all studies. In response, the authors compared both CDI and non-CDI diarrhea groups to healthy, which is great. However, they then go on to describe/justify the use of "non-IBD patients" from Gevers et al and Papa et al. as controls saying:

"We believe that including these analyses in our comparisons is valid even though the control populations are not strictly healthy. While not ideal, incorporating more studies increases our power to identify consistent disease-associated signals, and the increased power from including these two studies far outweighs the slight discrepancy in control groups."

I am not sure that I agree with this. If your control group is not healthy, the design is more suited for finding IBD specific microbial signatures rather than "core" disease associated changes. It is not clear in the paper what this "non-IBD patient group" specifically represents. What sort of illnesses did they have? Without this information it is hard to evaluate whether this is a "slight discrepancy in control groups" or a big discrepancy. It actually says on lines 707-708 (page 23) of the paper that the controls in Gevers et al are healthy.

2) Genus names in Figures 14 and 15 are too small to read.

3) For HIV, the authors note that the Noguera-Julian paper found sexual behavior (namely Men who have sex with men; MSM) to be a major confounding factor in HIV-associated microbiome studies. For the re-analysis of these data, were all "healthy" controls compared to all HIV positive individuals regardless of MSM status? If you really do want to focus on the effect of HIV disease versus health, these re-analyses should really be conducted such that sexual behavior is carefully controlled for, as was done by the Noguera-Julian paper. It is not clear in the manuscript whether this was done. The dataset of Lozupone et al is definitely driven largely by MSM and not HIV disease so I would not use any differences in there to help define core differences with disease.

4) With regards to my comment 4, that the definition of core was not very rigorous, the authors argue that they think that it is. I do not think Stouffer's method is a good way to define a core either. I agree with them that it could be driven by high significance in one study so am not impressed to see that their definition is more conservative than that. Although I agree that finding that the same genus differing in the same direction in 2 different diseases may be "worth noting" – I really don't think that finding the same genus twice across 28 different studies of 10 diseases convinces me that makes it a "core microbial response to health and disease" or supports statements like on lines 282-283 where they say "most previously reported microbe-disease associations may not be specific to individual diseases but instead likely reflect a universal microbial response to disease" – how does sporadic detection within studies of 2 of 10 diseases become "universal"? How many times would you find changes in the same dominant gut genera just by chance? This is especially the case since these analyses are conducted at the course genus level and the observation of a genus changing in two different diseases might not actually be the same thing (i.e. different species within that genus are driving the result in the different diseases).

Reviewer #3 (Remarks to the Author):

Overall the authors did an excellent job addressing the concerns of the reviewers and updating the manuscript to reflect those changes. The response to the authors was very rigorous and well written. This was clearly a lot of work and I do think the manuscript greatly improved with these revisions.

I noted one point for further clarification.

It appears there was some confusion regarding the first reviewer's point made at the bottom of page 7. As the authors pointed out, a model that randomly assigned values as being of one group or another would always predict an approximately 50:50 split of the data, regardless of how uneven the sample size was. But if, for example, a dataset was very uneven and 90% of the samples were "healthy", a model could be created to simply always state that the sample was "healthy" and it would be correct 90% of the time. Perhaps it is not a truly "random" model, but it could still achieve high accuracy without actually using any data. The kappa statistic the reviewer mentioned would correct for this type of imbalance throughout the datasets by accounting for both the observed and expected accuracy against such a null model as "always classifying as healthy".

It will be important for the authors to mention (one or two sentences) any potential problems that could have been caused by imbalanced sampling in the datasets. I also think the authors should include a supplemental table with the kappa values associated with the Figure 1 AUC values, so that the reader has a better idea of how this model performed when considering the data distribution.

Summary of the response

We thank the reviewers for their critical re-reading of our revised manuscript. As requested by the editor, we have addressed the reviewers' comments and made further revisions to the submitted manuscript.

Specifically, we:

- reframed our main text results to emphasize that many microbes may respond non-specifically to more than one disease, rather than focusing on presenting a “core” response to health and disease
- included discussion of the statistical and biological significance of the non-specific response in the supplement
- added discussion of the effect of class imbalances on our case vs. control classifiers to the main text and supplement

We respond to each reviewer point-by-point below, and have also included in our submission a manuscript file with all revisions tracked.

Reviewer #2

Reviewer #2 remained concerned that our heuristic definition was not rigorous enough to support our claim of identifying a “core response to health and disease.” They also suggested that in defining these “core” microbes, we should only consider datasets that had healthy controls and that were not demonstrably confounded by non-disease effects (e.g. behavior, as in the HIV Lozupone dataset).

We agree with the points raised by Reviewer #2 that our heuristic definition is not strong enough to be able to define a set of true “core” microbes, and have reframed our results to instead emphasize that many microbes may respond non-specifically to more than one disease. We recognize that the current amount of data presented is not expansive enough to establish which microbes comprise a core response to health and disease. Future meta-analyses, which include many more datasets for each of many conditions, may be able to develop more statistically

rigorous definitions to identify core microbes which are consistently associated with health or disease. We also agree with Reviewer #2's recommendation that only datasets with healthy controls be considered when defining the non-specific microbial response, and have adjusted our results accordingly.

I feel that the paper has been improved based on the responses to the Reviewer's comments. However, there were a couple of things in their rebuttal that I did not quite agree with:

1) Reviewer 1 made the point that in the re-analysis of the CDI data, that the non-CDI diarrheal controls were grouped with the healthy controls, and that they should confirm that for each disease group that the control samples are similar across all studies. In response, the authors compared both CDI and non-CDI diarrhea groups to healthy, which is great. However, they then go on to describe/justify the use of "non-IBD patients" from Gevers et al and Papa et al. as controls saying:

"We believe that including these analyses in our comparisons is valid even though the control populations are not strictly healthy. While not ideal, incorporating more studies increases our power to identify consistent disease-associated signals, and the increased power from including these two studies far outweighs the slight discrepancy in control groups."

I am not sure that I agree with this. If your control group is not healthy, the design is more suited for finding IBD specific microbial signatures rather than "core" disease associated changes. It is not clear in the paper what this "non-IBD patient group" specifically represents. What sort of illnesses did they have? Without this information it is hard to evaluate whether this is a "slight discrepancy in control groups" or a big discrepancy. It actually says on lines 707-708 (page 23) of the paper that the controls in Gevers et al are healthy.

- **Author response:** We agree that the non-IBD vs. IBD comparisons are more suited for finding IBD-specific microbial signatures rather than "core" disease-associated changes. We have removed the two IBD datasets which used non-healthy patients as controls (IBD Gevers and IBD Papa) from the non-specific microbial response definition.
- The specific illnesses that these control patients had are described in the original publications: in IBD Gevers this is described as "control subjects with non-inflammatory conditions—for example, presenting with abdominal pain and diarrhea" and in IBD Papa these are "patients with symptomatology suggestive of IBD: constipation (n = 9), abdominal pain (n = 8), gastroesophageal reflux (n = 2), poor weight gain (n = 1), diarrhea (n = 1), blood in stool (n = 2) and oropharyngeal dysphagia (n = 1)." We have clarified these control populations in the Supplementary Information (lines 733-735 and 761-763), and fixed the discrepancy on lines 707-708 (now line 733).

2) Genus names in Figures 14 and 15 are too small to read.

- **Author response:** We have increased the size of genus labels on Figures 14 and 15. The figure is saved in vector format, so readers should be able to zoom in and view the genus names. Furthermore, the raw data (with full taxonomies) is available in Supplementary Files S1 and S5 available at:
<https://github.com/cduvallet/microbiomeHD/tree/master/final/supp-files>

3) For HIV, the authors note that the Noguera-Julian paper found sexual behavior (namely Men who have sex with men; MSM) to be a major confounding factor in HIV-associated microbiome studies. For the re-analysis of these data, were all “healthy” controls compared to all HIV positive individuals regardless of MSM status? If you really do want to focus on the effect of HIV disease versus health, these re-analyses should really be conducted such that sexual behavior is carefully controlled for, as was done by the Noguera-Julian paper. It is not clear in the manuscript whether this was done. The dataset of Lozupone et al is definitely driven largely by MSM and not HIV disease so I would not use any differences in there to help define core differences with disease.

- **Author response:** As with the nonIBD vs. IBD comparisons mentioned above, we agree that including this dataset in defining non-specific responses to disease is not appropriate. Because HIV status in the Lozupone dataset is confounded with MSM status, we have removed it from the datasets we use in our heuristic, and have updated the Methods to clarify this accordingly (lines 504-507): “When we defined these non-specific genera, we did not include datasets which used non-healthy controls (Papa et al. (2012) and Gevers et al. (2014)) and the Lozupone et. al (2013) dataset, where the microbiome signal reflected behavior rather than disease state.”

4) With regards to my comment 4, that the definition of core was not very rigorous, the authors argue that they think that it is. I do not think Stouffer’s method is a good way to define a core either. I agree with them that it could be driven by high significance in one study so am not impressed to see that their definition is more conservative than that. Although I agree that finding that the same genus differing in the same direction in 2 different diseases may be “worth noting” – I really don’t think that finding the same genus twice across 28 different studies of 10 diseases convinces me that makes it a “core microbial response to health and disease” or supports statements like on lines 282-283 where they say “most previously reported microbe-disease associations may not be specific to individual diseases but instead likely reflect a universal microbial response to disease” – how does sporadic detection within studies of 2 of 10 diseases become “universal”? How many times would you find changes in the same dominant gut genera just by chance? This is especially the case since these analyses are conducted at the course genus level and the observation of a genus changing in two different diseases might not actually be the same thing (i.e. different species within that genus are driving the result in the different diseases).

- **Author response:** We investigated whether the number of non-specific responders we observed was greater than we would expect to see due to chance. When using significance in 2 diseases as the defining threshold, we did not find significantly more non-specific genera than would be expected by chance. However, as we increased the threshold to 3 and 4 diseases, the number of health-associated bacteria was significantly larger than would be expected by chance but much smaller than with the threshold of 2 diseases. Thus, there is currently not enough data to fully distinguish between microbes that are sporadically detected across multiple diseases from those that are consistently associated with general health or disease, but future studies which include more datasets and diseases may be able to find a coherent and robust response to health and disease. Although this analysis was not significant, our other results support the hypothesis of a non-specific, shared response (e.g. non-specific responders have a coherent phylogenetic signal; health vs. disease classifiers successfully classify case patients across many diseases). We discuss the statistical significance and biological

interpretation of these results further in the Supplementary Section 8.4 (lines 1082-1116).

- However, we do agree with the reviewer that labeling microbes which respond to multiple diseases as a “core” response to disease is likely too strong of a claim. We do not wish to distract from our main point, which is that certain genera respond non-specifically to disease. Therefore, we have re-framed the results in the main text accordingly (i.e. removing reference to a ‘core’ or ‘non-core’ response and replacing it with a ‘specific’ or ‘non-specific’ response to disease).

Reviewer #3

Reviewer #3’s remaining concern was related to how class imbalances could have affected our classification results. We have included a table of kappa values in the supplement, and commented on the potential problems caused by imbalanced classes in the main text.

Overall the authors did an excellent job addressing the concerns of the reviewers and updating the manuscript to reflect those changes. The response to the authors was very rigorous and well written. This was clearly a lot of work and I do think the manuscript greatly improved with these revisions.

I noted one point for further clarification.

It appears there was some confusion regarding the first reviewer’s point made at the bottom of page 7. As the authors pointed out, a model that randomly assigned values as being of one group or another would always predict an approximately 50:50 split of the data, regardless of how uneven the sample size was. But if, for example, a dataset was very uneven and 90% of the samples were “healthy”, a model could be created to simply always state that the sample was “healthy” and it would be correct 90% of the time. Perhaps it is not a truly “random” model, but it could still achieve high accuracy without actually using any data. The kappa statistic the reviewer mentioned would correct for this type of imbalance throughout the datasets by accounting for both the observed and expected accuracy against such a null model as “always classifying as healthy”.

It will be important for the authors to mention (one or two sentences) any potential problems that could have been caused by imbalanced sampling in the datasets. I also think the authors should include a supplemental table with the kappa values associated with the Figure 1 AUC values, so that the reader has a better idea of how this model performed when considering the data distribution.

- **Author response:** We originally chose to report classifier results with AUCs precisely because this metric is not affected by class imbalances (i.e. even with a 90/10 class split, a random classifier would report an AUC of 0.5). Other metrics like accuracy, kappa scores and fisher p-values are sensitive to class imbalances and can only measure performance for a classifier at one probability threshold cutoff. The ROC AUC measures performance by evaluating true and false positive rates at different probability thresholds. Unlike accuracy, TPR and FPR are not affected by different class distributions because they are each calculated based on only one of the classes (in other words, the ratio of the classes does not appear in the calculation for either TPR or FPR).
[doi:10.1016/j.patrec.2005.10.010].

- However, we agree that reporting additional performance metrics can be useful to help readers interpret these classifier results. We have included fisher's p-values and kappa scores for the classifiers in Supplementary Table 5. These scores were calculated on the results from scikit-learn's `RandomForestClassifier.predict()` function applied to the test data in each cross-validation fold. It should be noted that the `predict()` function uses a probability threshold of 0.5, and thus may not reflect the results from the optimal classifier for each dataset.
- We have also added the following sentence to the Methods section: "To account for spurious high classifiability due to class imbalances, we also calculated the Cohen's kappa score for each classifier using `sklearn.metrics.cohen_kappa_score` on the test set predictions. The kappa scores correlated well with the AUCs (Pearson $r = 0.9$), indicating that the majority of the classifiers performed well even when considering their underlying data distributions." and include Supplementary Table 5 with the kappa scores for each classifier.

REVIEWERS' COMMENTS:

Reviewer #2 (Remarks to the Author):

The authors adequately addressed all of my concerns in this revision.

Reviewer #3 (Remarks to the Author):

The authors did a great job addressing the reviewer concerns and I have no remaining points for revision.

I also want to reiterate that, in addition to the manuscript, the associated GitHub repository is very nicely done. The authors should be proud of their work here.